# Design, Development, and Validation of the Self-Perceived Health Scale (SPHS)

**DOI:** 10.3390/healthcare11142007

**Published:** 2023-07-12

**Authors:** Lorena Ishel Tinajero-Chávez, José Fernando Mora-Romo, Andrea Bravo-Doddoli, Beatríz Viridiana Cruz-Narciso, Nazira Calleja, Filiberto Toledano-Toledano

**Affiliations:** 1Facultad de Psicología, Universidad Nacional Autónoma de México, Ciudad Universitaria, Coyoacán, Mexico City 04510, Mexico; lorenaishel@gmail.com (L.I.T.-C.); andbrado@hotmail.com (A.B.-D.); ncalleja@unam.mx (N.C.); 2Facultad de Estudios Superiores Iztacala, Universidad Nacional Autónoma de México, Tlalnepantla 54090, Mexico; j_fmora@hotmail.com (J.F.M.-R.); bv.cruz.n72@gmail.com (B.V.C.-N.); 3Unidad de Investigación en Medicina Basada en Evidencias, Hospital Infantil de México Federico Gómez Instituto Nacional de Salud, Márquez 162, Cuauhtémoc, Mexico City 06720, Mexico; 4Unidad de Investigación Multidisciplinaria en Salud, Instituto Nacional de Rehabilitación Luis Guillermo Ibarra Ibarra, Calzada México-Xochimilco 289, Arenal de Guadalupe, Tlalpan, Mexico City 14389, Mexico; 5Dirección de Investigación y Diseminación del Conocimiento, Instituto Nacional de Ciencias e Innovación para la Formación de Comunidad Científica, INDEHUS, Periférico Sur 4860, Arenal de Guadalupe, Tlalpan, Mexico City 14389, Mexico

**Keywords:** self-perceived health, validation of psychological instruments, patient-reported outcomes (PRO), reliability, psychometrics, measurement, scales, constructs

## Abstract

Health is a multidimensional concept with notable psychological factors, such as self-perceived health (SPH). SPH is defined as the subjective assessment of individual health status, and it integrates information related to both physical and psychological aspects, such as lifestyle. This study describes the development of the Self-Perceived Health Scale (SPHS), and its validation in a Mexican sample (n = 600). Exploratory factor analysis (n = 303) and confirmatory factor analysis (n = 293) were carried out, and they supported the three-dimensionality of the SPH construct: physical health, psychological health, and healthy lifestyle. A final 12-item scale was obtained, and the scale showed adequate validity and reliability, as well as measurement invariance between sexes, indicating its robustness.

## 1. Introduction

Health is a multidimensional construct with various facets, including biological and social factors, and its investigation has been both extensive and intensive [1]. Psychological factors, such as self-perceived, self-assessed, self-reported, or subjective health, have also been considered, reflecting the elements related to the preservation, deterioration, or improvement of general health [1].

An assessment measurement of health is self-perceived health (SPH) which has been defined as the individual evaluation of personal health status, including physical and psychological health and the maintenance of a healthy lifestyle [2]. SPH has been recognized as a reliable and valid measure of actual health status from a subjective perspective, reporting important characteristics of health [3]. It is associated with epidemiological indicators of mortality and morbidity, as well as various biological markers such as blood pressure, lipid profile, and body mass index [4]. It has even been considered a fundamental pillar in the design of biological aging markers, such as the DunedinPACE aging clock [5].

SPH has been identified as a better predictor of individual mortality probability and functional capacity than chronic-degenerative diseases, as it considers the subjective assessment of the subtle effects of pathology that may go unnoticed in objective health assessments such as physical and chemical medical examinations [6]. Thus, the information provided by subjective health assessment allows for important decisions regarding the choice and design of medical treatments required by the patient, considering their potential impact on quality of life, where prioritization of those treatments that have a lesser effect on it are recommended [7,8]. In this sense, SPH has also been linked to biological dysregulations caused by stress in terms of allostatic load, which refers to the accumulation of negative effects due to chronic stress that induces harmful changes in biological function and increases the likelihood of developing disease [9]. It has shown consistency in its evaluation from adolescence to adulthood and the ability to predict the level of allostatic load in young adults, highlighting the need to modify habits from adolescence as a strategy for preventing morbidity in adulthood [4].

Lazarevic [10] describes four properties of SPH that justify its widespread use as a health indicator. Firstly, SPH is recognized as an integrative construct because respondents have the ability to converge different aspects when evaluating their own health, including manifest or known facts such as suffering from a chronic-degenerative disease, as well as more subjective aspects such as the development of specific symptoms that are independent of those evaluated in medical reviews or those related to a diagnosed illness. Secondly, SPH allows the incorporation of information about the observed health status at the time of assessment as well as possible short-term changes, which are essential for assessing overall health. The third property describes the integration of health-related habits in the evaluation, which can be modified when individuals perceive the capacity to improve their health, enhancing their motivation to maintain or adopt healthy habits. Finally, the evaluation of SPH is influenced by psychosocial aspects such as socioeconomic status, perceived social support, and overall psychological resources that individuals believe to have at their disposal to cope with health-related imbalances.

So, by SPH being a concept focusing on the subjective and individual evaluation of health status, through which different types of sensations and perceptions can be integrated, provides a precise information about physiological processes that may not be highlighted in clinical evaluations, such as inflammatory response [2,10]. It primarily focuses on assessing the overall health status, without explicitly considering the consequences of that state on different relevant aspects of a person’s life, such as their physical function or social role, which are related to well-being or health-related quality of life [11].

Gender differences have been identified in the assessment of SPH, which lead to unreliable results when relating it to mortality. According to Benyamini et al. [12], these differences are due to women incorporating different sources of information when self-evaluating their health compared to men, who rely solely on the severity of the disease, disregarding other non-specific factors to a life-threatening disease such as pain. Therefore, it is essential to propose a measurement that unifies the criteria for both men and women and specifically determines what respondents are considering when self-assessing their own health.

For this, it must be clearly determined what is being measured when using SPH as an indicator in studies of different countries, considering the factors that influence health perception and the elements on which respondents can base their evaluations. In this regard, it has been pointed out that physical condition is one of the determining factors of SPH, which is described as the ability to perform daily activities autonomously and without limitations [13]. Strongly associated with physical condition is Physical Health Status, understood as the extent to which an individual perceives themselves as healthy, with sufficient energy, and free from pain, as well as functionality and physical condition related to mobility, i.e., an individual’s capacity to move, maintain balance and coordination, and engage in intense, frequent, and sustained physical activity [2]. Similarly, both authors [2,13] indicate that a Healthy Lifestyle, meaning the extent to which a person perceives that their daily activities related to nutrition and exercise contribute to the maintenance or improvement of their health, exerts a significant influence on how people evaluate their own health. This because these behaviors are associated with the perception of good health, and changes in them may be related to a negative perception of health.

Psychological factors are also important in the evaluation of SPH, including emotional and cognitive aspects. It has been reported that the degree to which a person perceives being in touch with their emotions, the perception that their life is going well, and their ability to cope with stressful events in their life, are related to how they evaluate their own health [13]. While physical factors are important for SPH, the relevance of psychological factors had been highlighted for their important role in what people consider significant when evaluating their SPH [2].

Independently from well-being and quality of life, SPH transcends through its relationship with the outcomes of health prevention and care programs, showing an association with the absence of disease, the effectiveness of medical treatments received by patients, and a lower number of health complaints [14]. In the face of this situation, at the time of assessing SPH in studies, it is common to use only one question “*How is your health in general?*” with five-point response scale (excellent–poor), sometimes accompanied by another question related to the individual’s physical condition or their comparison of health with others of the same age. 

However, there is a lack of data on the validity and reliability of this measurement [2]. This kind of question usually is included in health assessment instruments such as the SF-36 Questionnaire [15], the Coop-Wonca Charts [16], and the Nottingham Health Profile [17], where self-perceived health is considered as part of the health status evaluation but does not encompass all dimensions of the construct. 

Due to the inclusion of SPH as part of screening instruments for health, psychological well-being, and health-related quality of life, confusion has arisen regarding both constructs. Although they are related, they identify different aspects of an individual’s health status. While SPH primarily focuses on the subjective evaluation that an individual makes by integrating various types of sensory and cognitive information, ultimately leading to their conclusion of how healthy they perceive themselves to be [2,10,11], health-related quality of life describes the individual’s perception of their position in life within a certain cultural context and under certain values, with respect to life goals and expectations, including social, cultural, and economic factors in its assessment. It also evaluates the impact of a person’s health status on different domains of their life, highlighting their ability to perform various functions that are important to them from both subjective and objective perspectives. As a result, SPH is considered as one dimension among others in this type of scale.

As mentioned, the assessment of self-perceived health is commonly carried out through several items incorporated into health screening instruments, social well-being, and health-related quality of life, among which the following stand out: (a) *36-Item Short Form Health Survey Questionnaire* (SF-36) [18], which is used to evaluate health-related quality of life through multiple dimensions related to different aspects of health, such as physical functioning, social functioning, role limitations due to emotional problems, mental health, vitality, bodily pain, and social well-being; (b) *Nottingham Health Profile* (NHP), used in primary care, designed to provide information about individual health problems, consists of six dimensions: physical ability, pain, emotional reactions, sleep, and social isolation, which focus on the impact of health on an individual’s daily functioning and overall quality of life; (c) *World Health Organization Quality of Life Brief* (WHOQOL-BREF), a brief instrument that evaluates overall quality of life, consisting of four dimensions: social relationships, environment, psychological health, and physical health. In the latter dimension, questions related to the perception of health status are included, such as the item “How satisfied are you with your health?” However, strictly speaking, it does not assess whether the individual considers their health to be good or bad; (d) *EuroQoL-5*, a health-related quality of life instrument based on five dimensions: mobility, self-care, daily activities, pain and discomfort, and anxiety and depression. It is widely used in national surveys as it provides an overall profile of health status [19].

In view of this conceptual and operational confusion, there is a need to be able to differentiate the construct of SPH from Health-Related Quality of Life, as the latter has been developed to assess those aspects of the individual’s subjective experience focused primarily on how illness, disability, and treatment itself impacts on people’s quality of life [20]. In contrast, the study of SPH has focused on identifying the functional physical status of individuals, regardless of whether any illness, disability, or ongoing treatment is present [21]. This becomes relevant since, as reported by Moon [20] in the absence of measurement instruments really focused on the assessment of SPH, the use of instruments designed to assess quality of life is considered appropriate, despite the differences that have been pointed out regarding the objectives of each instrument.

Despite SPH is a widely used health indicator in epidemiological research, public health, and social medicine, which has been considered equivalent to an individual’s latent health status [10], the attributions that a person makes to determine their health status are highly ambiguous [2]. Similarly, due to the inability to evaluate the psychometric properties of the measurement since it is assessed through a single question, it is unknown what respondents understand by self-perceived health when answering the question, whether all individuals understand the same thing, and if they consistently describe the same evaluation through their response.

Another issue related to the current measurements of SPH are the response options. The most common ones used are: *excellent, very good, good, fair, or poor*, although the wording tends to vary in different studies, making comparisons difficult. Similarly, it presents problems regarding symmetry between positive and negative options, as well as in translation into languages such as Spanish, where the difference between “good” and “fair” is subtle and confusing [22]. For this reason, it is important to consider the interpretation of the responses, as the options may have different meanings for the respondents. For example, when participants are asked to choose one of the options regarding their perception of their health through a single question, there is a possibility that they consider different elements in their evaluation. Additionally, single-item scales are not sensitive enough to detect small changes in the construct being evaluated.

One of the most significant negative consequences of evaluating the construct through a single item is the inability to estimate the reliability of this type of measurement. Internal consistency coefficients cannot be calculated, which is why it is suggested to use Visual Analogue Scales (VAS) in conjunction with other measurement instruments to improve reliability and validity in construct measurement [23].

Since SPH is usually used in national health surveys aimed at developing public policies to improve healthcare for citizens in different countries, due to the characteristics of the one question measurement, it becomes challenging to make valid and reliable comparisons between the results of different studies [22].

Although it has been noted that the construct is robust based on its correlations and its behavior in national health surveys in different countries, it is essential to create a measurement scale that integrates all its dimensions and based on the meanings provided by the respondents. This will objectively determine what they are referring to when responding about their health status.

For this, in the present research SPH was defined as the subjective evaluation of individual health status, incorporating different types of information related to self-perception of mobility, physical aspects of movement and balance, psychological well-being, and the assessment of one’s own behavior as healthy [2,10,11].

Since the psychometric properties cannot be assessed with single-item measures [24], the aim of this study was to develop a Self-Perceived Health scale that allows assess the different dimensions comprising the construct. It is essential for research purposes to have valid and reliable instruments that measure relevant health-related measures. The main contribution of this study is the creation of a reliable and valid scale, ensuring the comprehensive, sensitive, and invariant measurement of the construct according to gender. Thus, the obtained results can be valid for both men and women and comparable among populations with different backgrounds. In the first phase, the scale was developed considering seven distinct factors. The items were constructed based on feedback received regarding the dimensions proposed by Jylhä [2], Shields and Shooshtari [13], from participants during focus groups [25], and cognitive interviews [26]. The items wording was based on the guidelines proposed by DeVellis and Thorpe [27]. The validity was evaluated in the second phase, in which the factorial structure was identified through exploratory and confirmatory factor analysis, as well as reliability, gender invariance, and convergent validity assessed through two questions commonly used to evaluate self-perceived health: “How do you rate your general health status?” and “How do you rate your physical condition?” Construct validity was evaluated using the Brief Symptom Inventory-18 (BSI-18), the Coping Strategies Inventory COPE-28, and the Subjective Well-being Scale (EBS-8).

## 2. Materials and Methods

### 2.1. Phase I: Development of the Self-Perceived Health Scale

#### 2.1.1. Elaboration of the Scale

Based on a review of the literature and the responses obtained from two focus groups conducted with Mexican participants from the general population. The authors conducted the focus groups online via videocall with six people in each group. Seven questions were asked in three categories: physical health, psychological health, and healthy lifestyle. The reasoning behind determining Physical Health, Psychological Health, and Healthy Lifestyle as the main dimensions, and therefore conducting the focus groups based on these factors, was to provide conceptual coherence to the construct of this study since, as Moon [20] reports, frequently, although the aim of the study is to assess self-perceived health, measurement instruments and definitions derived directly from the Quality of Life construct are often used. Therefore, this decision was made based on the contributions of different authors who have worked on SPH [1,2,3,6,11].

Both sessions were videotaped with the prior informed consent of the participants and were subsequently transcribed textually in order to identify the meaning that the participants gave to the three aforementioned categories. The work of transcription and revision of the text was carried out by two co-authors and, in case of discrepancies, they were resolved by consensus of the other authors. To make sense of the transcribed text, a content analysis was performed where, among everything reported by the participants, meaning units that were related to the three central factors were first identified and labeled according to the factor to which they belonged in order to give greater structure to the information collected [28]. For example, some details of the material analyzed were phrases about the activities carried out by the participants that reflected adequate physical health such as “Frequency of mobility, not being static constantly. Being under constant movement”, “Balance with the way you self-regulate, cope with emotions, a loss, or some discontent around the day” regarding psychological health, and “having adequate habits at work and rest; considering that being every day working is not totally good, since relaxation and rest are part of health” referring to Healthy Lifestyle. This allowed us to identify the themes that described behaviors, experiences or emotions experienced by the participants with respect to the questions asked, offering a more detailed description that allowed us to made 52 items and expand the number of factors to consider those that Shields and Shooshtari [13], and Jylhä [2] recommended: (1) physical health, i.e., the degree to which a person perceives himself or herself to be healthy, energetic, and pain-free; (2) healthy lifestyle, i.e., the degree to which a person perceives that his or her daily activities, diet, and exercise allow him or her to preserve and/or improve his or her health; (3) physical mobility, i.e., the degree to which a person perceives himself or herself to be able to move, coordinate, and balance; (4) physical condition, i.e., the degree to which a person perceives that he or she can perform intense, frequent, and long-term physical activity; (5) mental health, i.e., the degree to which a person perceives that his or her behaviors favor his or her mental health by promoting well-being, calmness, and coping with stress; (6) emotional health, i.e., the degree to which a person perceives that being in contact with one’s emotions gives one control over oneself and one’s well-being; and (7) cognitive health, i.e., the degree to which a person perceives that his or her cognitive abilities, memory, learning, and execution of tasks are adequate. The contributions of these authors [2] regarding the factors were used since, to date, they have been the ones who have explored in greater depth the most relevant factors at the time of studying SPH.

#### 2.1.2. Content Validity

The 52 items were evaluated by seven experts in medicine and psychology. Once each of the evaluations had been obtained, the consistency between the judges was calculated using Aiken’s *V* index [29]. The objective of conducting this expert evaluation was to assess the degree of relevance among the items to support the proposed conceptual definition, and the degree to which the number of items was sufficient to adequately represent their respective factor. To this end, the experts who agreed to participate were given an e-mail form asking them to evaluate on a scale of 1 (Not at all relevant) to 5 (Fully relevant) each of the items; and on a scale of 1 (Not at all represented) to 5 (Fully represented) the degree to which the items were sufficient to support their respective factors. This was carried out as we sought to ensure that the final items were sufficiently consistent with the proposed conceptual definition, while achieving adequate conceptual coverage of their factors [30]. The formula used to calculate the Aiken’s V index was $V=\frac{\overset{-}{X}-l}{k}$ where $\overset{-}{X}$ is the mean of the judges’ evaluation scores, l is the lowest score that is possible to obtain, and k is the difference between the highest and lowest score on the rating scale [29,30]. The database with the experts’ evaluations can be accessed at the following link https://doi.org/10.17632/8wrysjbsny.1 (accessed on 19 June 2023).

The coefficients obtained for both the items and the dimensions were satisfactory, as they ranged between 0.85 (95% CI: 0.66–0.92) and 0.96 (95% CI: 0.80–0.97), with the exception of the items “I feel good about myself” (*V* = 0.38, 95% CI = 0.12–0.45) and “I feel mentally healthy” (*V* = 0.46, 95% CI = 0.27–0.62). These items were removed, yielding a scale with 50 items.

#### 2.1.3. Answer Options

Six answer options on an asymmetric Likert-type scale were used for respondents to indicate their level of agreement with each item, where 1 was “Disagree”, 2 “Slightly agree”, 3 “Agree”, 4 “Mostly agree”, 5 “Strongly agree”, and 6 “Absolutely agree”. This format was used to increase the variability in the distribution of the data and prevent a ceiling effect, which is essential for psychometric procedures [31]. A six-point Likert scale, without a neutral point, was chosen to prevent the intermediate response bias [32] where people tend to use the intermediate option to avoid providing a compromised response to the items, which would affect the consistency of the participants’ responses and consequently affect the performance of the exploratory and confirmatory factor analysis models. Additionally, we used asymmetric Likert-type response options because they are recommended when the intention is to measure variables where people tend to overestimate the perceptions that they have of themselves, causing a ceiling or floor effect [31]. Those authors have reported that using these types of response options allows researchers to obtain a wider variability in the participants’ data, thus minimizing the skew and kurtosis statistics and allowing researchers to obtain a univariate and multivariate normal distribution, which are important elements for the proper analysis of the data.

For the case of self-perceived health, we consider it appropriate to use this response format since, in the context of the Mexican population, it has been reported that this population tends to make an erroneous self-perception of their own health due to the optimistic overestimation of health status [33].

#### 2.1.4. Pilot Study

Five structured cognitive interviews were performed to evaluate comprehension of the sentence structure, item meanings, and answer format of the scale and to adjust the wording of the items as required. The cognitive interviews were conducted face-to-face following the recommendations of Willis [26], where one interview was conducted using the “Thinking aloud” technique, asking the participant to verbalize everything that came to mind while answering the scale. The second interview was conducted using a “Concurrent Paraphrasing” technique where the participant was asked to say the items in his or her own words as he or she answered the scale. The third interview was conducted using a “Retrospective paraphrasing” technique, where the participant, once he/she had completed the scale, was asked to state as many of the items as he/she could remember. The fourth interview was conducted by means of “exhaustive paraphrasing”, where the participant was asked to restate in his/her own words all the items of the scale (items, response options, and instructions). The fifth interview was conducted using the “Selective paraphrasing” technique, where the participant rephrased in his/her own words those items that, while answering the scale, he/she considered the most confusing. The objective of these interviews was to identify if there were items, answer options or instructions that were unfamiliar to the participants. Since the interviewees correctly paraphrased all the items on the scale, there was no need to modify them. 

Subsequently, a pilot study of the scale was performed with 50 participants with characteristics similar to those of the final sample to evaluate the clarity and understandability of each of the items, as well as to determine the distribution of answers based on the asymmetric response scale. The distribution obtained from the application of the 50 participants since their skewness and kurtosis were <|2|, the means obtained (lower $\overset{-}{X}$ = 3.08, upper $\overset{-}{X}$ = 5.06) were close to the theoretical mean ($\overset{-}{X}$ = 3.5, for six response options), and at the time of concluding the application, no participant reported having incomprehensible items when asked about the clarity of the scale.

### 2.2. Phase II: Validation of the Self-Perceived Health Scale

#### 2.2.1. Design

The study was instrumental [34] because the psychometric properties of the developed psychological instrument were analyzed.

#### 2.2.2. Participants

Purposive nonprobability sampling was used to recruit a sample of 600 Mexican participants from the general population; the data were divided into two databases. The first (n = 303) was used to carry out item discrimination analysis, exploratory factor analysis (EFA), and construct and convergent validity tests, while the second (n = 297) was used to perform confirmatory factor analysis (CFA) and tests of measurement invariance. The sample size was determined based on current recommendations for factor analysis procedures that indicate that they should be performed with two samples of participants [35]: one sample specifically for the exploratory factor analysis and the other for the confirmatory factor analysis. Likewise, it has been recommended that for conducting the EFA, a sample >300 participants should be employed, since the models perform more efficiently with this sample size. Upon considering that the factor loadings of the items in the EFA were >0.700, the second sample size for CFA could be conformed with a minimum of 200 participants. Overall, of the 600 participants, 68.8% were women; their ages ranged from 18 to 64 years (M = 31.37, SD = 9.48), and the majority had a graduate education (79%). Regarding their marital status, 61.3% were single, 21.3% were married, 12.2% were in a civil union, and 5.2% were in an open relationship.

#### 2.2.3. Instruments

In addition to the SPHS, which was formed during the first phase of the study, and a questionnaire to collect the sociodemographic data of the participants, a set of instruments was administered to obtain evidence of the construct validity of the SPHS: the Subjective Well-Being Scale, the Brief Coping Inventory, the Brief Symptom Inventory, and two questions to assess the state of general health and physical condition. Each instrument is described below.

*Sociodemographic variables questionnaire (Q-SV) for research in family caregivers of children with chronic diseases* [36]. This questionnaire contains 20 items that evaluate information on sociodemographic, medical, sociocultural, and family variables in families of children with chronic diseases. For this study, the diagnosis, age, and sex of the patient and caregiver, the relationship between the two (mother, father, or another family member), the educational level (no schooling, primary education, secondary education, undergraduate education, postgraduate education), occupation (housemaker, worker, trader, employee, student, pensioner, unemployed), marital status (married, living together, separated, divorced, single parent, widowed), years of partnership, number of children, type of family (nuclear, semi-nuclear, extended, single-parent), family life cycle (with young children, with school-age children, with adult children), social support networks (family, friends, religion, institutions, government), religion (Catholic, Christian, none), and monthly income were determined.

*Self-Perceived Health Scale (SPHS-50):* This scale evaluates individuals’ perception of their physical health, including their mobility and physical condition, their perception of their lifestyle as healthy, and their psychological health, considering both emotional and cognitive health. The scale comprises 50 items with six answer options in an asymmetric Likert format ranging from “Disagree” to “Absolutely agree”. Higher scores indicate a positive assessment of health.

*Subjective Well-Being Scale (SWBS-8)* [37]: This scale evaluates life satisfaction (LS) and positive affect (PA). The short version includes eight items with six asymmetric answer options; in the original study, α = 0.948 was reported for the LS subscale, α = 0.964 for the PA subscale, and α = 0.970 for the total scale. In the present study, we found an α = 0.961 for the total scale, an α = 0.927 for the PA subscale and an α = 0.954 for the LS subscale. According to the nomological network of the SPH construct, a positive association would be expected with the SWBS due to the correlations indicated in the literature with scales such as the Perceived Well-Being Scale [13], as well as the fact that they are sometimes confused as the same construct when being evaluated by the same instruments. We used the SWBS-8 due to its length and the fact that it was designed for the Mexican population. The fit index for this scale were CMIN/DF = 2.45, CFI = 0.996, GFI = 0.993 and RMSEA = 0.046.

*Brief Coping Inventory (COPE-28)* [38]: This instrument consists of 28 items that evaluate coping strategies when individuals are faced with stress in 14 dimensions: emotional support, social support, active coping, planning, substance use, humor, religion, self-distraction, denial, venting, self-blame, disengagement, positive reframing, and acceptance. It features six symmetric answer options to assess frequency, ranging from “I didn’t do this at all” to “I did this all the time”. The original study did not report the alpha coefficient of the total scale, but the coefficients of its dimensions varied between α = 0.71 and α = 0.80. In the present study, an α = 0.802 was obtained for the total scale, and since the dimensions of each factor comprised only 2 items, their alpha coefficients were not calculated. The fit index for this scale were CMIN/DF = 1.699, CFI = 0.957, GFI = 0.906 and RMSEA = 0.048.

According to the nomological network of the SPH construct, it would be expected that SPH would not be associated with coping because coping strategies have not been previously related to SPH in the literature; in addition, coping is defined as any effort to manage stress, and is therefore not directly related to the subjective perception of health.

*Brief Symptom Inventory (BSI-18)* [39]: This instrument evaluates the experience of depressive, anxious, and physical symptoms that require psychological attention. It has eighteen items with five symmetrical answer options ranging from “Not at all” to “A lot”. The inventory was validated in the Mexican population, with α = 0.839 reported for the depression factor, α = 0.784 for anxiety and α = 0.722 for somatization. In the present study, α = 0.905 for depression, α = 0.886 for anxiety, α = 0.833 for somatization, and for the total scale, an α = 0.938 was obtained. The fit index for this scale were CMIN/DF = 1.13, CFI = 0.996, GFI = 0.978 and RMSEA = 0.026.

According to the nomological network of the SPH construct, negative associations with the SPHS would be expected because the BSI-18 assesses the degree of discomfort and psychological distress experienced by individuals through anxious, depressive, and somatic symptomatology, it is expected that those individuals who experience a higher degree of distress perceive their health as being more deteriorated compared to that of those who experience fewer distress symptoms.

To assess the convergent validity of the SPHS and due to the lack of another psychological instrument measuring the construct, two questions that are frequently used to assess SPH [2] were applied: “*How do you rate your general health?*” and “*How do you rate your physical condition?*” There were six answer options ranging from “very poor” to “excellent”.

### 2.3. Ethical Considerations

This study is a part of the research project HIM/2015/017/SSA.1207 “Effects of mindfulness training on psychological distress and quality of life of the family caregiver”, which was approved by the Research, Ethics and Biosafety Commissions of the Hospital Infantil de México Federico Gómez National Institute of Health in Mexico City. While conducting this study, we followed the ethical rules and considerations for research with humans currently enforced in Mexico [40] and those outlined by the American Psychological Association [41]. All family caregivers were informed of the objectives and scope of the research and their rights according to the Helsinki Declaration [42]. The caregivers who agreed to participate in the study signed an informed consent letter. Participation in this study was voluntary and did not involve payment.

### 2.4. Data Analysis

The calculations were made using the statistical software SPSS version 26, AMOS version 24 and R-Studio program version 1.3.1093.

To determine the item discrimination, it was verified that the percentage of the answer option with the highest frequency was not >50%; that the mean obtained was as close to the theoretical mean according to the number of answer options proposed (mean = 3.5 for the 6 options), as well as the standard deviation, asymmetry and kurtosis (both <|3|); and that a corrected homogeneity index (cHI) greater than 0.30 and differences between extreme groups were obtained. This last parameter was obtained through quartiles 1 and 3 of the total score of the SPHS-50, and the participants were divided into three groups (low, medium, and high) based on the distribution of their responses to compare the scores of the low and high groups using Student’s *t*-test. A *p* value > 0.05 indicated that an item did not discriminate between extreme groups, and therefore, these items were considered for removal.

To analyze the factorial structure of the SPHS, the adequacy of the data was calculated to perform exploratory factor analysis (EFA) using the Kaiser–Meyer–Olkin (KMO) test and Bartlett’s test of sphericity, with the results being considered adequate with KMO > 0.80 and *p* < 0.05, respectively. Likewise, the anti-image correlation matrix was considered (*r* > 0.800). Parallel analysis was performed to determine the appropriate number of scale factors using the “*paran*” package in *R*. EFA was carried out using the *fa* function of the “*psych*” package. Due to compliance with univariate normality, the presence of more than five answer options and a sample of >300 participants, the maximum likelihood estimation method was chosen. Since the factors were expected to be correlated, oblique rotation was chosen. Cutoff points of 0.50 and 0.40 were established for communalities and for factorial loadings in one factor, respectively.

To confirm multivariate normality, the Mardia test was performed, which confirms the assumption of multivariate normality if the multivariate kurtosis coefficient is less than p(p+2), where p is the number of variables observed [43].

The tests with the CFA models were carried out using the maximum likelihood estimation method. To evaluate the degree of fit of the models, the following parameters were used: chi-square statistic (*X*^2^), comparative fit index (CFI, expected value >0.90), and root mean square error of approximation (RMSEA, expected value ≤0.08) with its corresponding confidence interval [44]. The average variance extracted (AVE) was calculated using the factorial loadings (λ) obtained from the CFA, where an adequate convergent internal validity is considered if the AVE value is >0.50 [45].

Reliability was calculated using Cronbach’s alpha coefficient and the omega coefficient for the total scale and for each of the estimated factors, considering a coefficient >0.80 to be adequate, in addition to the alpha coefficient if the item was removed.

Measurement invariance between genders (women and men) was evaluated by comparing a model with unconstrained parameters (configurational or baseline model) with three different models: (1) the metric model, called the weak model, with constrained factor loadings; (2) the scalar, or strong, model, where factor loadings and intercepts were constrained; and (3) the strict model, where factor loadings, intercepts, and error variances were constrained. The statistics used to consider the existence of measurement invariance were the differences between the chi-square scores (Δχ^2^) and the changes in CFI and RMSEA. Strong invariance is detected if the differences between the CFI values are ≤0.001, if the RMSEA is ≤0.0015 and if the Δχ^2^ results in a *p* > 0.05 [46]. Pearson’s correlation tests were used to evaluate convergent, divergent, and construct validity.

## 3. Results

### 3.1. Item Discrimination Analysis

None of the 50 items had to be removed due to item discrimination procedures based on the frequency distribution, asymmetry, kurtosis, and differences between extreme groups or corrected homogeneity indices (cHI). The item with the highest percentage for an answer option was item 1 “*I consider my physical health to be adequate*”, at 30%. The item with the mean that was farthest from the theoretical mean was item 4, “*I can make coordinated movements*” (M = 5.02). In general, the standard deviations were homogeneous among the items, ranging from 1.20 to 1.70, while the cHI values ranged from 0.487 to 0.827. Finally, all the items had significant values in the Student’s t test for extreme groups (*p* < 0.001).

### 3.2. Exploratory Factor Analysis

The KMO obtained with the 50 items was 0.97, while the anti-image correlation ranged from 0.94 to 0.98; Bartlett’s test of sphericity yielded χ^2^(1225) = 15476.72, *p* = 0.001; therefore, the data were considered adequate for performing the EFA.

The parallel analysis suggested extracting three factors for the EFA since three variables with percentages of variance higher than the 95th percentile were obtained [47].

Because various items were found with factor loadings <0.40 (items 11, 15, 22, 25, 29, and 47), as well as with communalities <0.50 (items 1, 14, 26, 35, 46, and 48), the decision was made to remove these items one by one while performing a new parallel analysis and EFA each time, resulting in a scale of 38 items. To obtain a short version, the items with the highest factor loadings were chosen to perform model comparisons using scales with 38, 30, 24, 18, and 12 items. The outputs from these alternatives scale version are shown in Table 1.

Table 2 shows the EFA results for the 12-item scale: factor loadings of the items, communalities, Alpha if the item is removed and total explained variance.

Table 3 shows the reliability for the total scale (SPHS-12) and its factors (Psychological Health, Physical Health, and Healthy Lifestyle), the interfactor correlations and the means and standard deviations.

### 3.3. Confirmatory Factor Analysis

For this analysis, a new sample of participants (n = 297) whose data were not included in the EFA was used. For this second sample, another group of participants who had not been involved in the first sample was selected. The reason for this was that, rather than seeking equivalence in the sample characteristics—such as age or sex—what was intended was to confirm that the EFA results of the first sample would be stable in the application of a second sample totally independent of the first sample and achieve adequate results in the CFA [35]. It is also important to note that the reason the first group was not given the additional measurement instruments was because they were of no practical use during the EFA stage. However, since it was necessary to establish construct validity during the CFA stage, they were applied to the second sample of participants.

Having confirmed the multivariate normality by means of the multivariate kurtosis coefficient (Kurtosis = 78.885), which was lower than that calculated by Bollen’s [43] formula, 12×(12+2)=168, we carried out the CFA. For the CFA, in addition to the 12-item scale, four alternative models with 38, 30, 24, and 18 items distributed among the three factors presented were compared. The sample used for these alternative models was the same sample used for the EFA (n = 303). Each model consisted of the same three factors: psychological health (PsH), physical health (PhH), and lifestyle (LS).

As shown in Table 4, the model with 12 items had the best fit indices, so it was decided to use these items to form the final scale.

The AVE obtained for the SPHS-12 was 0.751, a value higher than the cutoff point of 0.50 proposed by Fornell and Larcker [45]. Figure 1 shows the structural model together with the correlation and standardized regression coefficients and the variance of each item.

### 3.4. Measurement Invariance

Measurement invariance by gender (men/women) was evaluated using the “*Lavaan*” package from *R*. The unconstrained model (configural invariance) was compared with a model with constrained factor loadings (metric invariance); later, this model was compared with a model with factor loadings and constrained intercepts (scalar invariance) to finally compare the latter with a model in which the residuals were also constrained (strict invariance). The results of these comparisons are found in Table 5.

It was not possible to obtain strict invariance; however, given the discussions of Bentler [48], who reported that strict invariance is excessively restrictive, the partial invariance presented by the SPHS-12 was considered adequate since scalar invariance suggests that the differences in the factor means between groups indicates a true group difference [46,49].

Therefore, in the Student’s *t* test for independent samples between men and women, significant differences were found for the psychological health factor (M_women_ = 3.65, S.D._women_ = 1.28; M_men_ = 4.01, S.D._men_ = 1.39, *t* = −2.119, df = 295, *p* = 0.035), physical health factor (M_women_ = 5.03, S.D._women_ = 1.12; M_men_ = 5.49, S.D._men_ = 0.826, *t* = −3.429, df = 295, *p* = 0.001) and the total score of the SPHS-12 (M_women_ = 4.11, S.D._women_ = 1.05; M_men_ = 4.48, S.D._men_ = 0.989, *t* = −2.78, df = 295, *p* = 0.006). No significant differences were observed between genders for the lifestyle factor.

### 3.5. Construct Validity

Again, the first database (n = 303) was used for these analyses. To obtain evidence of construct validity for the SPHS-12, correlations were obtained with the scores of scales that evaluate constructs proposed by the SPH nomological network (Table 6).

Regarding the Subjective Well-Being Scale (SWBS-8), we expected to obtain positive correlations, an approach that was corroborated by the moderate correlations between the SWBS-8 and the following factors: psychological health (r = 0.688, *p* = 0.001), healthy lifestyle (*r* = 0.510, *p* = 0.001), and the total SPHS-12 (*r* = 0.617, *p* = 0.001); the correlation with the physical health factor (*r* = 0.325, *p* = 0.001) was weak [50].

The relationship of the SPHS with the Brief Symptom Inventory (BSI-18) was expected to be negative, which was confirmed through the correlations between the SPHS factors psychological health (*r* = −0.595, *p* = 0.001), physical health (*r* = −0.130, *p* = 0.024), and healthy lifestyle (*r* = −0.369, *p* = 0.001) and the SPHS-12 total scale (*r* = −0.499, *p* = 0.001) with the Brief Symptom Inventory.

Last, it was suggested that SPH would not be related to coping strategies evaluated using the COPE-28 scale, which showed weak correlations with psychological health (*r* = 0.334, *p* = 0.001), physical health (*r* = 0.252, *p* = 0.001), healthy lifestyle (*r* = 0.300, *p* = 0.001) and the full scale (*r* = 0.356, *p* = 0.001). Based on these data, it is suggested that the construct validity of the SPHS-12 was adequate for convergent (SWBS-8) and divergent (BSI-18) validity. However, the discriminant validity of the SPHS-12 was barely sufficient given that, although the correlations between the COPE-28 and SPHS-12 were the lowest when compared to the BSI-18 and SWBS-8, the correlation between COPE-28 and lifestyle was higher than between BSI-18 and lifestyle.

### 3.6. Convergent Validity

Convergent validity was evaluated by correlating the SPHS-12 scores with those of two general questions: “How do you rate your general health?” (M = 3.87, SD = 0.99) and “How do you rate your physical condition?” (M = 3.64, SD = 1.25). The values for these questions correlated positively and significantly with the three factors, namely, psychological health (*r* = 0.596, *p* = 0.001; *r* = 0.365, *p* = 0.001), physical health (*r* = 0.471, *p* = 0.001; *r* = 0.474, *p* = 0.001), and healthy lifestyle (*r* = 0.642, *p* = 0.001; *r* = 0.652, *p* = 0.001), as well as with the total SPHS-12 (*r* = 0.686, *p* = 0.001; *r* = 0.594, *p* = 0.001).

## 4. Discussion

The results obtained from the development and validation of the SPHS-12 can be considered of great value since, to date, the authors of this study had not been able to find another psychometric scale focused exclusively on the assessment of self-perceived health. As mentioned at the beginning of this paper, when assessing this construct, a single general question is usually used (“How good do you consider your health to be?” [18]), or questionnaires whose objectives are to assess other variables such as quality of life [15], which has led some authors to consider this variable as an imprecise measurement, despite its importance in the estimation of population mortality [51].

Although the Current Perceived Health-42 Questionnaire [52] was previously published, it cannot be considered as an SPH measurement instrument either, since the objective of this instrument was to assess the current state of health of the participants, which was considered by the authors as a different and independent variable from SPH. Thus, the SPHS-12 is probably the first scale specifically designed and validated to assess Self-Perceived Health in the general population.

Another advantage of this scale is its trifactorial configuration (Physical Health, Psychological Health, and Lifestyle), which allows for more specific assessments of what domains participants consider “Healthy”, as opposed to other attempts to assess this perception through general questions that did not allow discerning whether what participants considered “healthy” was their cognitive capacity, their physical mobility, or their diet, to mention some aspects that are addressed in the items of the scale presented.

Through the evaluation of the dimensionality of the construct through EFA and parallel analysis, it was observed that the suggested model presented only three factors, a solution that was maintained by removing items that lacked the appropriate psychometric characteristics, which yielded a scale comprising 38 items. Because SPH is commonly included in brief general health evaluation instruments, we chose to develop a brief scale; hence, it was decided to test its dimensionality with a minimum number of items (4 per dimension) and verify whether the suggested structure was sustainable, which was confirmed by selecting the 12 items of the final scale, i.e., those that presented the highest factor loadings.

With respect to the dimensions identified by the scale and due to the differences indicated in the literature regarding what the professionals define and what the respondents understand regarding SPH [10], it was decided to determine first-hand what people understand as health; it is interesting to note that through the construction of the scale, it can be seen that the participants do not consider their cognitive functioning in the evaluation of self-perceived health, giving greater importance to physical and emotional health and lifestyle.

Moreover, it was possible to identify that people integrate their physical condition and mobility into general physical health, so that when asking about their general health status, as suggested by single-question assessments, respondents may be answering information only related to their ability to move and not with all the dimensions that make up the general health status.

Subsequently, a CFA of the scales with 38 and 12 items was carried out; both scales showed a good fit, with all the items belonging to each dimension having high factor loadings. In the same way, both scales presented satisfactory reliability indices and exhibited invariance by gender. This may indicate that when the three dimensions of the construct are contemplated, that is, the construct is fully evaluated, the differences indicated in the literature with respect to gender are diluted, resulting in an equivalent factorial structure both in men and women.

Likewise, the discrepancies with other authors regarding the differences in the SPH between men and women [6] may be due to the age of the sample, since although the ages of respondents ranged from 18 to 64 years, the mean was 31.37 years, and some authors, such as Fielding and Li [52], have pointed out that it is after the age of 40 that the differences in the perception of one’s own health appear between men and women.

The results relating the SPHS-12 with the specified nomological network were satisfactory. It was verified that the SPHS-12 was positively related to subjective well-being, although it was highlighted that the physical health dimension showed a weak correlation with the SWBS-8, suggesting that it is necessary to differentiate between well-being and physical health, which are constructs that are sometimes evaluated through a single measurement (see, for example, [15]). This finding becomes relevant based on the tests that confuse the measurement of self-perceived health and well-being [11,53], since, as can be seen through the results of the developed scale, the respondents attribute their physical health to their ability to move regardless of their emotional health, which is more related to well-being.

Regarding the relationship with the measurement of depressive, anxious, and physical symptoms (BSI-18), it was verified that the SPHS-12 was strongly and negatively related to the experience of discomfort, unease, and somatization symptoms, both globally and in the dimensions of psychological health and healthy lifestyle. Meanwhile, the correlation with the physical health factor, although negative, was weak, so it is possible that what the respondents understood as the evaluation of SPH was less related to the characteristics of physical health. This finding is very important because it indicates that the ability to move and travel, that is, the physical health dimension of the SPHS-12, based on what the participants understand as self-perceived health, is poorly related to the Global Severity Index provided by the BSI-18, which has been consistently related to the diagnosis of various chronic-degenerative diseases such as cancer [39].

## 5. Suggestions for Future Research

For future research, it is suggested that the measurement invariance analysis be performed more in depth to verify that the factorial structure is constant in populations with different sociodemographic characteristics, such as age, place of residence, educational level, and presence of chronic or acute diseases.

Similarly, future works could opt to try to assess the criterion validity of SPHS-12 using biomarkers that reflect autonomic nervous system activity, such as peripheral temperature, Heart Rate Variability (HRV), respiratory rate, or inflammatory response using variables such as Interleukin-6 (IL-6), C-reactive protein, or Tumor Necrosis Factor alpha (TNF-α). The reasons for proposing these biomarkers are that recent studies have reported interesting associations where irregular values of these biomarkers (e.g., low peripheral temperature, high levels of IL-6, or sympathetic-dominant HRV) produce a low assessment of Self-perceived Health [54].

In this sense, future research could also evaluate the feasibility of using the SPHS-12 as an assessment tool in future psychological intervention protocols focused on fostering appropriate autonomic regulation through strategies such as mindfulness, diaphragmatic breathing, guided imagery, and progressive muscle relaxation [55,56].

## 6. Conclusions

The results of the validation process of the SPHS support the multidimensionality of the construct [1]; by means of factorial analyses, a structure with three factors, i.e., physical health, psychological health, and healthy lifestyle, was identified and confirmed.

The SPHS-12 was shown to be a valid and reliable scale to assess HPS in the Mexican population, supporting a robust factorial structure, which can be used in both men and women at different vital stages.

The dimensions identified through the dimensional analysis of the SPHS-12 show that the construct has an important psychological and behavioral component through psychological health and lifestyle, which showed that the respondents place great value on the behavioral repertoire through the which they take care of their own health.

Likewise, the physical health dimension was shown to be related to the ability to move their extremities, walk and maintain balance, more than with the perception of a good physical condition, the absence of disease or the experience of symptoms of distress.

Through the development of the SPHS-12, we not only created a multidimensional instrument to measure SPH but also clearly determined what is truly being evaluated [16]. The results indicate that subjectively assessed health is largely related to subjects’ cognitive evaluation of their actions to improve or preserve their health (the healthy lifestyle factor) through their interpretations of perceived discomfort or unease, which do not necessarily have a real basis at the time of physical assessment but show a potential to predict disease onset or deterioration.

## Figures and Tables

**Figure 1 healthcare-11-02007-f001:**
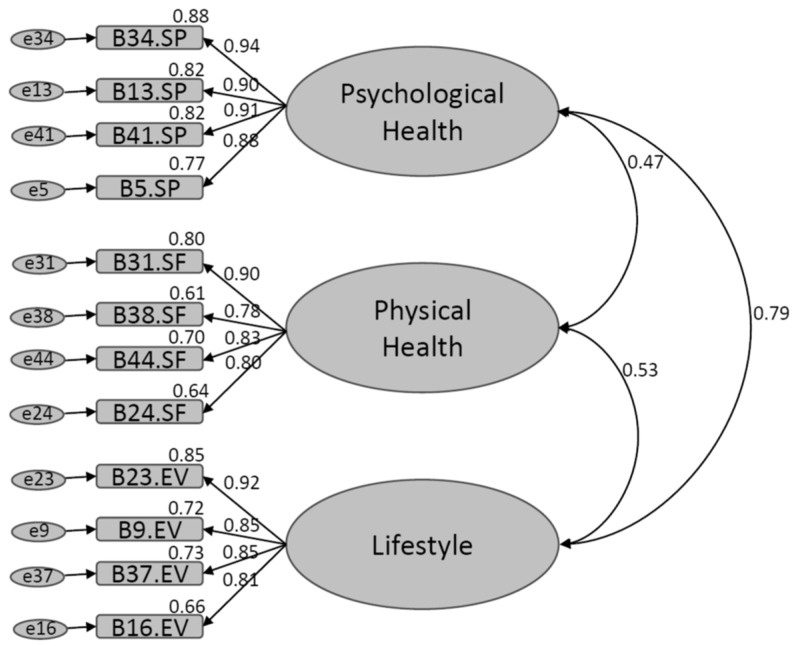
Structural model of the SPHS-12.

**Table 1 healthcare-11-02007-t001:** Factorial loading, distribution and communalities from the successive versions obtained.

50 Items SPHS				
Items	Psychological Health	Physical Health	Healthy Lifestyle	Communalities (H^2^)
13. I am emotionally healthy.	**0.899**	−0.131	0.105	0.808
34. I feel emotionally healthy.	**0.891**	−0.172	0.165	0.828
41. I consider myself a person with emotional well-being.	**0.886**	−0.131	0.142	0.827
5. I have an adequate level of mental health.	**0.816**	−0.121	0.143	0.711
45. I believe I am able to regulate my emotions.	**0.809**	0.055	0.005	0.707
6. I feel I am open about my feelings and emotions.	**0.773**	−0.043	−0.015	0.555
20. I accept my emotions and feelings.	**0.770**	−0.054	0.036	0.583
27. I am able to solve my problems appropriately.	**0.763**	0.086	0.064	0.719
33. I have the skills necessary to promote my own mental health.	**0.760**	0.037	0.090	0.694
50. I am aware of my emotions and feelings.	**0.721**	0.082	−0.071	0.529
40. My mental health makes it easier for me to do my daily activities.	**0.709**	0.119	0.094	0.694
7. I consider myself mentally agile.	**0.700**	0.111	−0.015	0.568
28. My mental capacity is adequate.	**0.676**	0.246	−0.027	0.657
12. I am able to cope with stressful situations in my life.	**0.667**	0.073	0.110	0.599
21. I consider my brain to be functioning properly.	**0.642**	0.287	−0.071	0.614
42. I can learn new things easily.	**0.635**	0.347	−0.198	0.575
49. I am able to finish my duties without a problem.	**0.600**	0.104	0.165	0.585
14. I have a good memory.	**0.587**	0.224	−0.147	0.420
48. I consider myself able to seek professional psychological help when I need it.	**0.586**	0.025	−0.071	0.316
46. I am able to remember details from the previous day.	**0.578**	0.289	−0.122	0.487
35. I am able to remember my daily commitments.	**0.576**	0.294	−0.071	0.526
26. Talking about my problems gives me peace of mind.	**0.510**	0.095	−0.029	0.299
22. I have enough energy to face the day.	**0.391**	0.276	0.258	0.587
29. I feel energetic enough for the rest of the day.	**0.366**	0.274	0.314	0.623
47. The time I sleep allows me to recover to start a new day.	**0.303**	0.123	0.276	0.347
38. I can move my limbs (arms, legs, and/or head) without effort.	0.114	**0.839**	−0.114	0.712
31. I am able to walk with ease.	0.122	**0.817**	−0.038	0.747
24. I am able to go up and down stairs with ease.	−0.017	**0.729**	0.233	0.743
44. I am able to maintain my body balance, e.g., standing on one foot.	0.247	**0.706**	−0.079	0.661
32. I feel able to do sport or physical exercise regularly.	−0.146	**0.684**	0.344	0.694
10. I am able to move easily.	0.161	**0.669**	0.145	0.727
4. I can make coordinated movements (e.g., stand on one foot, walk on tiptoe, etc.).	0.214	**0.646**	−0.041	0.567
39. I consider myself capable of physical exercise for at least 30 min.	−0.092	**0.633**	0.326	0.636
3. My ability to move around and walk is adequate.	0.094	**0.588**	0.173	0.562
17. I can perform my daily activities with ease.	0.190	**0.567**	0.160	0.618
11. I am able to perform intense physical activity.	−0.162	**0.565**	0.436	0.619
18. I have physical strength.	−0.011	**0.564**	0.337	0.616
25. I consider myself physically agile.	−0.004	**0.512**	0.441	0.685
15. I am free of pain.	0.297	**0.298**	0.207	0.439
9. My habits allow me to maintain my health.	0.055	0.006	**0.856**	0.793
2. I feel my lifestyle is healthy.	0.133	−0.087	**0.794**	0.688
23. My habits help me maintain my health.	0.144	0.028	**0.779**	0.778
16. I feel that my daily activities improve my health.	0.054	0.072	**0.751**	0.675
37. My diet helps me maintain good health.	0.142	0.035	**0.749**	0.731
30. I do activities that help me improve my health.	0.053	0.087	**0.740**	0.672
8. Physically I feel healthy.	0.186	0.154	**0.609**	0.676
43. The exercise and/or physical activity I do helps me maintain my health.	−0.049	0.293	**0.594**	0.573
19. My daily activities make me feel good.	0.205	0.114	**0.593**	0.632
36. I consider my physical health to be adequate.	0.151	0.100	**0.556**	0.505
36. I feel healthy.	0.348	0.190	**0.480**	0.730
Number of items: 50	25	14	11	
Total explained variance: 62.68%	28.17%	17.56%	16.95%	
**38 items SPHS**				
**Items**	**Psychological Health**	**Physical Health**	**Healthy Lifestyle**	**Communalities** **(H^2^)**
13. I am emotionally healthy.	**0.938**	−0.094	0.038	0.835
34. I feel emotionally healthy.	**0.919**	−0.143	0.112	0.844
41. I consider myself an emotionally healthy person.	**0.909**	−0.096	0.093	0.843
5. I have adequate mental health.	**0.841**	−0.089	0.089	0.724
45. I believe I am able to regulate my emotions.	**0.821**	0.084	−0.035	0.719
20. I accept my emotions and feelings.	**0.772**	−0.022	0.006	0.584
6. I feel I am open to my feelings and emotions.	**0.761**	−0.018	−0.027	0.543
33. I have the necessary skills to promote my own mental health.	**0.760**	0.063	0.067	0.700
27. I am able to solve my problems adequately.	**0.753**	0.122	0.038	0.718
40. My mental health makes it easier for me to do my daily activities.	**0.701**	0.141	0.075	0.694
12. I am able to cope with stressful situations in my life.	**0.692**	0.092	0.058	0.610
50. I am aware of my emotions and feelings.	**0.691**	0.098	−0.063	0.504
7. I consider myself mentally agile.	**0.669**	0.131	−0.008	0.549
28. My mental capacity is adequate.	**0.647**	0.276	−0.033	0.647
21. I consider my brain to function adequately.	**0.607**	0.316	−0.070	0.599
42. I can learn new things easily.	**0.575**	0.362	−0.163	0.530
49. I can finish my duties without a problem.	**0.554**	0.117	0.183	0.564
38. I can move my limbs (arms, legs, and/or head) without effort.	0.054	**0.878**	−0.109	0.725
31. I can walk with ease.	0.071	**0.867**	−0.047	0.775
24. I am able to go up and down stairs with ease. 24.	−0.055	**0.739**	0.238	0.741
44. I am able to maintain my body balance, e.g., by standing on one foot.	0.190	**0.729**	−0.060	0.655
10. I can move with ease.	0.115	**0.702**	0.140	0.737
4. I can make coordinated movements (e.g., standing on one foot, walking on tiptoe, etc.).	0.178	**0.684**	−0.048	0.583
32. I feel able to do sports or physical exercise on a regular basis.	−0.148	**0.674**	0.325	0.661
39. I consider myself able to exercise for at least 30 min.	−0.093	**0.628**	0.304	0.611
3. My ability to move around and walk is adequate.	0.054	**0.625**	0.166	0.582
17. I can perform my daily activities with ease.	0.134	**0.573**	0.183	0.604
18. I have physical strength.	−0.017	**0.550**	0.325	0.587
9. My habits help me maintain my health.	0.020	−0.012	**0.890**	0.802
23. My habits help me maintain my health.	0.092	0.007	**0.837**	0.806
16. I feel that my daily activities improve my health.	0.004	0.047	**0.806**	0.699
2. I feel that my lifestyle is healthy.	0.122	−0.090	**0.791**	0.672
30. I engage in activities that help me improve my health.	0.015	0.063	**0.787**	0.692
37. My diet allows me to maintain good health.	0.101	0.026	**0.784**	0.744
19. My daily activities make me feel good.	0.161	0.107	**0.631**	0.647
43. The exercise and/or physical activity I do helps me to maintain my health.	−0.079	0.280	**0.624**	0.584
8. I feel physically healthy.	0.188	0.157	**0.577**	0.650
36. I feel healthy.	0.341	0.197	**0.454**	0.712
Number of items: 38	17	11	10	
Total explained variance: 67.03%	29.09%	19.33%	18.61%	
**30 Items SPHS**				
**Items**	**Psychological Health**	**Healthy Lifestyle**	**Physical Health**	**Communalities** **(H^2^)**
13. I am emotionally healthy.	**0.945**	−0.009	−0.033	0.852
34. I feel emotionally healthy.	**0.930**	0.071	−0.092	0.864
41. I consider myself an emotionally healthy person.	**0.912**	0.58	−0.045	0.856
5. I have adequate mental health.	**0.850**	0.049	−0.042	0.738
45. I believe I am able to regulate my emotions.	**0.801**	−0.056	0.135	0.711
20. I accept my emotions and feelings.	**0.751**	−0.007	0.022	0.575
33. I have the skills necessary to foster my own mental health.	**0.740**	0.053	0.14	0.694
6. I feel that I am open to my feelings and emotions.	**0.738**	−0.039	0.028	0.532
27. I am able to solve my problems adequately.	**0.711**	0.037	0.163	0.692
40. My mental health makes it easier for me to do my daily activities.	**0.683**	0.056	0.181	0.687
9. My habits allow me to maintain my health.	0.007	**0.905**	−0.025	0.800
23. My habits help me to maintain my health.	0.061	**0.873**	−0.013	0.816
16. I feel that my daily activities improve my health.	−0.011	**0.831**	0.023	0.703
30. I engage in activities that help me improve my health.	−0.007	**0.815**	0.041	0.698
37. My diet helps me maintain good health.	0.077	**0.805**	0.015	0.745
2. I feel that my lifestyle is healthy.	0.116	**0.797**	−0.096	0.667
19. My daily activities make me feel good.	0.138	**0.652**	0.094	0.649
43. The exercise and/or physical activity I do helps me to maintain my health.	−0.092	**634**	0.265	0.583
8. I feel physically healthy.	0.182	**0.568**	0.162	0.647
36. I feel healthy.	0.334	**0.440**	0.213	0.712
31. I can walk easily.	0.052	−0.085	**0.914**	0.797
38. I can move my limbs (arms, legs, and/or head) without effort.	0.026	−0.132	**0.911**	0.727
44. I am able to maintain my body balance, for example, when standing on one foot.	0.149	−0.059	**0.750**	0.639
24. I am able to go up and down stairs with ease.	−0.079	0.232	**0.746**	0.737
10. I am able to move easily.	0.095	0.113	**0.737**	0.748
4. I can make coordinated movements (e.g., standing on one foot, walking on tiptoe, etc.).	0.160	−0.072	**0.716**	0.587
32. I feel able to do sports or physical exercise on a regular basis.	−0.142	0.299	**0.670**	0.647
3. My ability to move around and walk is adequate.	0.049	0.131	**0.655**	0.592
39. I consider myself capable of exercising for at least 30 min.	−0.086	0.272	**0.635**	0.605
17. I can do my daily activities with ease.	0.103	0.184	**0.578**	0.590
Number of items: 30	10	10	10	
Total explained variance: 69.63%	25.15%	22.61%	21.87%	
**24 Items SPHS**				
**Items**	**Psychological Health**	**Healthy Lifestyle**	**Physical Health**	**Communalities** **(H^2^)**
13. I am emotionally healthy.	**0.943**	−0.023	−0.002	0.861
34. I feel emotionally healthy.	**0.931**	0.058	−0.066	0.874
41. I consider myself a person with emotional well-being.	**0.891**	0.062	−0.022	0.843
5. I have adequate mental health.	**0.843**	0.039	−0.016	0.739
45. I believe I am able to regulate my emotions.	**0.775**	−0.035	0.0149	0.701
20. I accept my emotions and feelings.	**0.732**	0.006	0.040	0.571
6. I feel I am open about my feelings and emotions.	**0.730**	−0.037	0.052	0.540
33. I have the skills necessary to promote my own mental health.	**0.724**	0.064	0.122	0.696
23. My habits help me maintain my health.	0.49	**0.889**	−0.017	0.827
9. My habits allow me to maintain my health.	0.019	**0.876**	−0.014	0.772
16. I feel that my daily activities improve my health.	−0.014	**0.851**	0.013	0.722
30. I do activities that help me improve my health.	−0.014	**0.849**	0.023	0.730
37. My eating habits allow me to maintain good health.	0.074	**0.798**	0.018	0.731
2. I feel that my lifestyle is healthy.	0.133	**0.764**	−0.084	0.645
19. My daily activities give me well-being.	0.114	**0.689**	0.087	0.670
43. The exercise and/or physical activity I do helps me maintain my health.	−0.087	**0.658**	0.237	0.592
31. I can walk with ease.	0.032	−0.062	**0.920**	0.811
38. I can move my limbs (arms, legs and/or head) without effort.	0.007	−0.104	**0.903**	0.723
44. I can maintain my body balance, for example, by standing on one foot.	0.126	−0.030	**0.746**	0.636
10. I can move easily.	0.086	0.122	**0.742**	0.754
24. I am able to go up and down stairs with ease.	−0.090	0.248	**0.737**	0.733
4. I can make coordinated movements (e.g., standing on one foot, walking on tiptoe, etc.).	0.146	−0.053	**0.724**	0.599
3. My ability to move around and walk is adequate.	0.052	0.133	**0.656**	0.594
32. I feel able to engage in sport or physical exercise on a regular basis.	−0.130	0.316	**0.630**	0.614
Number of items: 24	8	8	8	
Total explained variance: 70.74%	24.61%	24.25%	21.87%	
**18 Items SPHS**				
**Items**	**Psychological Health**	**Healthy Lifestyle**	**Physical Health**	**Communalities** **(H^2^)**
34. I feel emotionally healthy.	**0.938**	0.041	−0.057	0.877
13. I have adequate emotional health.	**0.938**	−0.018	0.004	0.863
41. I consider myself a person with emotional well-being.	**0.890**	0.052	−0.005	0.846
5. I have adequate mental health.	**0.843**	0.040	−0.013	0.743
45. I believe I am able to regulate my emotions.	**0.774**	−0.053	0.167	0.698
20. I accept my emotions and feelings.	**0.711**	0.005	0.054	0.551
9. My habits allow me to maintain my health.	−0.016	**0.914**	−0.007	0.811
23. My habits help me maintain my health.	0.032	**0.869**	0.022	0.812
37. My eating allows me to maintain good health.	0.333	**0.836**	0.033	0.770
16. I feel that my daily activities improve my health.	−0.010	**0.802**	0.047	0.678
2. I feel that my lifestyle is healthy.	0.106	**0.801**	−0.082	0.682
30. I do activities that help me improve my health.	−0.004	**0.798**	0.047	0.677
31. I can walk with ease.	0.009	−0.035	**0.922**	0.824
38. I can move my limbs (arms, legs, and/or head) without effort.	−0.013	−0.073	**0.901**	0.733
44. I am able to maintain my body balance, for example, by standing on one foot.	0.108	−0.020	**0.760**	0.652
24. I am able to go up and down stairs with ease.	−0.124	0.290	**0.733**	0.738
10. I can move easily.	0.064	0.168	**0.713**	0.731
4. I can make coordinated movements (e.g., standing on one foot, walking on tiptoe, etc.).	0.138	−0.033	**0.710**	0.590
Number of items: 18	8	8	8	
Total explained variance: 73.75%	25.75%	25.35%	22.66%	

Source: Elaborated by the authors.

**Table 2 healthcare-11-02007-t002:** Factorial analysis of the SPHS−12.

		Factors (*λ*)				
Items		Psychological Health	Physical Health	Healthy Lifestyle	Communalities (H^2^)	Alpha If the Item is Removed
34. I feel emotionally healthy. [34. Me siento emocionalmente saludable.]		**0.944**	0.012	−0.023	0.885	0.926
13. I am in adequate emotional health. [13. Tengo una salud emocional adecuada.]		**0.940**	−0.036	0.027	0.686	0.926
41. I consider myself to be a person with emotional well-being. [41. Me considero una persona con bienestar emocional.]		**0.864**	0.060	0.019	0.829	0.925
5. I have adequate mental health. [5. Tengo una adecuada salud mental.]		**0.856**	0.020	0.001	0.755	0.927
31. I can walk easily. [31. Puedo caminar con facilidad.]		0.020	**0.904**	−0.002	0.838	0.929
38. I can move my limbs (arms, legs, and/or head) without effort. [38. Puedo mover mis extremidades (brazos, piernas y/o cabeza) sin esfuerzo.]		0.010	**0.899**	−0.026	0.794	0.931
44. I can keep my balance, for example, by standing on one foot. [44. Puedo mantener el equilibrio de mi cuerpo, por ejemplo, al pararme en un pie.]		0.051	**0.825**	0.025	0.759	0.930
24. I can go up and down stairs with ease. [24. Soy capaz de subir y bajar escaleras con facilidad.]		0.014	**0.784**	0.036	0.661	0.928
23. My lifestyle helps me take care of my health. [23. Mi estilo de vida me ayuda a cuidar mi salud.]		0.036	−0.047	**0.928**	0.845	0.925
9. My habits allow me to improve my health. [9. Mis hábitos me permiten mejorar mi salud.]		0.003	−0.057	**0.886**	0.734	0.926
37. My diet allows me to maintain good health. [37. Mi alimentación me permite mantener una buena salud.]		0.110	0.010	**0.734**	0.635	0.925
16. I feel that my daily activities improve my health. [16. Siento que mis actividades diarias mejoran mi salud.]		−0.105	0.309	**0.705**	0.736	0.928
Number of items	Total 12	4	4	4		
Total explained variance (%)	77.82%	28.07%	26.41%	23.34%		

**Table 3 healthcare-11-02007-t003:** Reliability, interfactor correlations and mean (SD) of the SPHS-12.

	Total	Factors		
		Psychological Health	Physical Health	Healthy Lifestyle
Reliability	α = 0.933 ω = 0.925	α = 0.951 ω = 0.952	α = 0.907 ω = 0.908	α = 0.926 ω = 0.927
Interfactor correlations	Psychological health	1		
	Physical health	0.463	1	
	Healthy lifestyle	0.605	0.558	1
Mean (SD)	3.98 (1.07)	3.60 (1.35)	3.49 (1.32)	4.87 (1.18)

**Table 4 healthcare-11-02007-t004:** Model fit indices.

Models	χ^2/^gl = CMIN <3	CFI >0.95	GFI >0.95	TLI >0.95	RMSEA (CI 90%)
SPHS-38 (n = 303)	2448.693/703 = 3.69	0.849	0.668	0.839	0.094 (0.090, 0.098)
SPHS-30 (n = 303)	1535.364/402 = 3.81	0.879	0.731	0.869	0.096 (0.091, 0.102)
SPHS-24 (n = 303)	903.262/249 = 3.62	0.909	0.786	0.900	0.093 (0.087, 0.100)
SPHS-18 (n = 303)	411.926/132 = 3.12	0.947	0.865	0.939	0.084 (0.075, 0.093)
SPHS-12 (n = 303)	126.407/51 = 2.479	0.978	0.936	0.971	0.070 (0.055, 0.085)
SPHS-12 (n = 297)	137.132/51 = 2.689	0.973	0.926	0.965	0.076 (0.060, 0.091)

**Table 5 healthcare-11-02007-t005:** Measurement invariance by gender.

**Model**	**X^2^(gl)**	**X^2^/gl**	**CFI**	**RMSEA (CI 90%)**		**ΔX^2^**	**ΔCFI**	**ΔRMSEA**
M1. Configuration invariance	219.253(102)	2.149	0.963	0.088 (0.072, 0.104)				
M2. Measurement or weak invariance (constrained λ)	230.996(111)	2.081	0.962	0.085 (0.070, 0.101)	M2 Vs M1	11.743 (9), *p* = 0.228	−0.001	−0.003
M3. Scalar invariance (constrained λ and τ)	240.864(120)	2.007	0.962	0.082 (0.067, 0.097)	M3 Vs M2	9.868 (9), *p* = 0.361	0.000	−0.003
M4. Strict invariance (constrained λ, τ and θ)	277.416(132)	2.101	0.954	0.086 (0.072, 0.100)	M4 Vs M3	36.553 (12), *p* = 0.0002	−0.008	0.004
Criteria						*p* > 0.05	≤0.01	≤0.015

**Table 6 healthcare-11-02007-t006:** Correlations between the total scores of the measurement scales.

	Psychological Health	Physical Health	Lifestyle	SPHS-12
SWBS-8	0.688 **	0.510 **	0.325 **	0.617 **
BSI-18	−0.595 **	−0.369 **	−0.130 *	−0.449 **
COPE-28	0.334 **	0.300 **	0.252 **	0.356 **

* *p* = 0.05, ** *p* = 0.001. *SPHS-12*: Self-Perceived Health Scale. *SWBS-8*: Subjective Well-Being Scale. *BSI-18*: Brief Symptom Inventory. *COPE-28*: Brief Coping Inventory.

## Data Availability

The raw data supporting the conclusions of this article will be made available by the authors without undue reservation.

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
