# Peer review of "Design, Development, and Validation of the Self-Perceived Health Scale (SPHS)"

_healthcare, 2023, doi:10.3390/healthcare11142007_

Round 1

Reviewer 1 Report

I would like to applaud the Author/s for exploring this topic, which is of particular interest to the current practice of keeping healthy and also to speak skillfully about it. However, to reach its full potential, the article needs to be edited; to provide further evidence and arguments to support its relevance, to provide further clarification on the selection and characteristics of the two research samples, to provide clear and concise scale development results, clear conclusions, managerial and theoretical implications.

I have strong doubts about the theoretical part of the article. It is only contained in the Introduction. The Introduction section needs a lot of attention in terms of putting the problem in the foreground. As it stands, the author does not justify the relevance well. I suggest that the Introduction section be reworded and that a theoretical background section be added. This should be done in the following steps:

The introduction should provide a reasonable rationale for the study and reflect its significance:

- In other words, why is this study important?

- How does it differ from previous studies?

- How can the study contribute to the literature?

Add a Theoretical Background section, as there is a weak theoretical foundation in the current form.

- Better explain the concept of Self-perceived health and its different dimensions.

- It was not specified why the factors recommended for measuring SPH by Shields and Shooshtari and Jylhä were used in the selection of questionnaire items? This should be included in the theoretical section of the article.

- In this section, a more detailed presentation of the rationale for the SPHS and its components is recommended.

Although the article explores an interesting and important topic, the theoretical gaps and the theoretical basis for creating the scales are very limited. The introduction does not clearly indicate the gap and the need for the SPH scale. There is also a lack of systematisation of the clear differences and limitations of current approaches as well as current scales.

The statistical analysis is adequate for the study. Too little detail is given on the choice of research samples and their justification, and it is not clear how many ranks the Likert scales had.

References are adequate for the study, but are very poor. The literature cited is not recent. There are no direct references to previous research on the topic and no measures or scales constructed to date.

The Discussion section is quite limited. It needs to be significantly improved to highlight the results and their significance by explaining more clearly new or valuable points and results compared to previous studies. This section needs to include and discuss the results of the study with comments and support in agreement or disagreement with previous studies.

Very laconically described Suggestions for Future Research section. Research limitations, theoretical and practical implications are missing.

Good luck with your work!

Author Response

Comments and Suggestions for Authors

I would like to applaud the Author/s for exploring this topic, which is of particular interest to the current practice of keeping healthy and also to speak skillfully about it. However, to reach its full potential, the article needs to be edited; to provide further evidence and arguments to support its relevance, to provide further clarification on the selection and characteristics of the two research samples, to provide clear and concise scale development results, clear conclusions, managerial and theoretical implications.

I have strong doubts about the theoretical part of the article. It is only contained in the Introduction. The Introduction section needs a lot of attention in terms of putting the problem in the foreground. As it stands, the author does not justify the relevance well. I suggest that the Introduction section be reworded and that a theoretical background section be added. This should be done in the following steps:

The introduction should provide a reasonable rationale for the study and reflect its significance:

- In other words, why is this study important?

Dear reviewer, thanks to your observation the importance of the study has been remarked throughout the introduction, for example in lines 50-60:

“the information provided by subjective health assessment allows for important decisions regarding the choice and design of medical treatments required by the patient, considering their potential impact on quality of life, prioritizing those treatments that have a lesser effect on it [7,8]. In this sense, SPH has also been linked to biological dysregulations caused by stress in terms of allostatic load, which refers to the accumulation of negative effects due to chronic stress that induces harmful changes in biological function and increases the likelihood of developing disease [9]. It has shown consistency in its evaluation from adolescence to adulthood and the ability to predict the level of allostatic load in young adults, highlighting the need to modify habits from adolescence as a strategy for preventing morbidity in adulthood [4].”

And lines 204-211:

“Since the psychometric properties cannot be assessed with single-item measures [24], the aim of this study was to develop a Self-Perceived Health scale that allows assess the different dimensions comprising the construct. It is essential for research purposes to have valid and reliable instruments that measure relevant health-related measures. The main contribution of this study is the creation of a reliable and valid scale, ensuring the comprehensive, sensitive, and invariant measurement of the construct according to gender. Thus, the obtained results can be valid for both men and women and comparable among populations with different backgrounds”

- How does it differ from previous studies?

In order to answer your valuable question, we have pointed out the conceptual and operational differences that we bring with our study to clearly differentiate between self-perceived health and quality of life. To this end, we first describe some measurement instruments that have been erroneously used to operationalize self-perceived health, and then describe the conceptual differences between the two variables. This information can be found in lines 115-166 in the manuscript, which we reproduce below:

“In the face of this situation, at the time of assessing SPH in studies, it is common to use only one question "How is your health in general?" with five-point response scale (excellent - poor), sometimes accompanied by another question related to the individual's physical condition or their comparison of health with others of the same age.

However, there is a lack of data on the validity and reliability of this measurement [2]. This kind of question usually is included in health assessment instruments such as the SF-36 Questionnaire [15], the Coop-Wonca Charts [16], and the Nottingham Health Profile [17], where self-perceived health is considered as part of the health status evaluation but does not encompass all dimensions of the construct.

Due to the inclusion of SPH as part of screening instruments for health, psychological well-being, and health-related quality of life, confusion has arisen regarding both constructs. Although they are related, they identify different aspects of an individual's health status. While SPH primarily focuses on the subjective evaluation that an individual makes by integrating various types of sensory and cognitive information, ultimately leading to their conclusion of how healthy they perceive themselves to be [2,10,11], health-related quality of life describes the individual's perception of their position in life within a certain cultural context and under certain values, with respect to life goals and expectations, including social, cultural, and economic factors in its assessment. It also evaluates the impact of a person's health status on different domains of their life, highlighting their ability to perform various functions that are important to them from both subjective and objective perspectives. As a result, SPH is considered as one dimension among others in this type of scale.

As mentioned, the assessment of self-perceived health is commonly carried out through several items incorporated into health screening instruments, social well-being, and health-related quality of life, among which the following stand out: (a) 36-Item Short Form Health Survey Questionnaire (SF-36) [18], which is used to evaluate health-related quality of life through multiple dimensions related to different aspects of health, such as physical functioning, social functioning, role limitations due to emotional problems, mental health, vitality, bodily pain, and social well-being; (b) Nottingham Health Profile (NHP), used in primary care, designed to provide information about individual health problems, consists of six dimensions: physical ability, pain, emotional reactions, sleep, and social isolation, which focus on the impact of health on an individual's daily functioning and overall quality of life; (c) World Health Organization Quality of Life Brief (WHOQOL-BREF), a brief instrument that evaluates overall quality of life, consisting of four dimensions: social relationships, environment, psychological health, and physical health. In the latter dimension, questions related to the perception of health status are included, such as the item "How satisfied are you with your health?" However, strictly speaking, it does not assess whether the individual considers their health to be good or bad; (d) EuroQoL-5, a health-related quality of life instrument based on five dimensions: mobility, self-care, daily activities, pain and discomfort, and anxiety and depression. It is widely used in national surveys as it provides an overall profile of health status [19].

In view of this conceptual and operational confusion, there is a need to be able to differentiate the construct of SPH from Health-Related Quality of Life, as the latter has been developed to assess those aspects of the individual's subjective experience focused primarily on how illness, disability, and treatment itself impacts on people's quality of life [20]. In contrast, the study of SPH has focused on identifying the functional physical status of individuals, regardless of whether any illness, disability, or ongoing treatment is present [21]. This becomes relevant since, as reported by Moon [20] in the absence of measurement instruments really focused on the assessment of SPH, the use of instruments designed to assess quality of life is considered appropriate, despite the differences that have been pointed out regarding the objectives of each instrument.”

- How can the study contribute to the literature?

We hope to be able to meet your concern about the contributions of our study to the field we are addressing. To this end, we have added lines 207-211:

“The main contribution of this study is the creation of a reliable and valid scale, ensuring the comprehensive, sensitive, and invariant measurement of the construct according to gender. Thus, the obtained results can be valid for both men and women and comparable among populations with different backgrounds.”

Add a Theoretical Background section, as there is a weak theoretical foundation in the current form.

- Better explain the concept of Self-perceived health and its different dimensions.

A better explanation of the concept of self-perceived health is provided. To do so, we begin by mentioning the factors that make it up in lines 33-37:

“Health is a multidimensional construct with various facets, including biological and social factors, and its investigation has been both extensive and intensive [1]. Psychological factors, such as self-perceived, self-assessed, self-reported or subjective health, have also been considered, reflecting the elements related to the preservation, deterioration, or improvement of general health [1].”

This is followed by a more detailed explanation in lines 61-81:

“Lazarevic [10] describes four properties of SPH that justify its widespread use as a health indicator. Firstly, SPH is recognized as an integrative construct because respondents have the ability to converge different aspects when evaluating their own health, including manifest or known facts such as suffering from a chronic-degenerative disease, as well as more subjective aspects such as the development of specific symptoms that are independent of those evaluated in medical reviews or those related to a diagnosed illness. Secondly, SPH allows the incorporation of information about the observed health status at the time of assessment as well as possible short-term changes, which are essential for assessing overall health. The third property describes the integration of health-related habits in the evaluation, which can be modified when individuals perceive the capacity to improve their health, enhancing their motivation to maintain or adopt healthy habits. Finally, the evaluation of SPH is influenced by psychosocial aspects such as socioeconomic status, perceived social support, and overall psychological resources that individuals believe to have at their disposal to cope with health-related imbalances.

So, by SPH being a concept focusing on the subjective and individual evaluation of health status, through which different types of sensations and perceptions can be integrated, provides a precise information about physiological processes that may not be highlighted in clinical evaluations, such as inflammatory response [2,10]. It primarily focuses on assessing the overall health status, without explicitly considering the consequences of that state on different relevant aspects of a person's life, such as their physical function or social role, which are related to well-being or health-related quality of life [11].”

- It was not specified why the factors recommended for measuring SPH by Shields and Shooshtari and Jylhä were used in the selection of questionnaire items? This should be included in the theoretical section of the article.

We hope that we have adequately complied with your valuable observation on the following lines 253-268:

“…allowed us to made 52 items and expand the number of factors to consider those that Shields & Shooshtari [13], and Jylhä [2] recommended: 1) physical health, i.e., the degree to which a person perceives himself or herself to be healthy, energetic and pain-free; 2) healthy lifestyle, i.e., the degree to which a person perceives that his or her daily activities, diet and exercise allow him or her to preserve and/or improve his or her health; 3) physical mobility, i.e., the degree to which a person perceives himself or herself to be able to move, coordinate and balance; 4) physical condition, i.e., the degree to which a person perceives that he or she can perform intense, frequent and long-term physical activity; 5) mental health, i.e., the degree to which a person perceives that his or her behaviors favor his or her mental health by promoting well-being, calmness, and coping with stress; 6) emotional health, i.e., the degree to which a person perceives that being in contact with one’s emotions gives one control over oneself and one’s well-being; and 7) cognitive health, i.e., the degree to which a person perceives that his or her cognitive abilities, memory, learning and execution of tasks are adequate. The contributions of these authors [2] regarding the factors were used since, to date, they have been the ones who have explored in greater depth the most relevant factors at the time of studying SPH.”

Later on, we described again the importance of considering the dimensions reported by several authors, including Shields and Shooshtari, and Jylhä on lines 228-235:

 “The reasoning behind determining Physical Health, Psychological Health and Healthy Lifestyle as the main dimensions, and therefore conducting the focus groups based on these factors, was to provide conceptual coherence to the construct of this study since, as Moon [20] reports, frequently, although the aim of the study is to assess Self-perceived Health, measurement instruments and definitions derived directly from the Quality of Life construct are often used. Therefore, this decision was made based on the contributions of different authors who have worked on SPH [1,2,3,6,11].”

- In this section, a more detailed presentation of the rationale for the SPHS and its components is recommended.

We again hope that this point has been correctly answered by lines 61-81:

“Lazarevic [10] describes four properties of SPH that justify its widespread use as a health indicator. Firstly, SPH is recognized as an integrative construct because respondents have the ability to converge different aspects when evaluating their own health, including manifest or known facts such as suffering from a chronic-degenerative disease, as well as more subjective aspects such as the development of specific symptoms that are independent of those evaluated in medical reviews or those related to a diagnosed illness. Secondly, SPH allows the incorporation of information about the observed health status at the time of assessment as well as possible short-term changes, which are essential for assessing overall health. The third property describes the integration of health-related habits in the evaluation, which can be modified when individuals perceive the capacity to improve their health, enhancing their motivation to maintain or adopt healthy habits. Finally, the evaluation of SPH is influenced by psychosocial aspects such as socioeconomic status, perceived social support, and overall psychological resources that individuals believe to have at their disposal to cope with health-related imbalances.

So, by SPH being a concept focusing on the subjective and individual evaluation of health status, through which different types of sensations and perceptions can be integrated, provides a precise information about physiological processes that may not be highlighted in clinical evaluations, such as inflammatory response [2,10]. It primarily focuses on assessing the overall health status, without explicitly considering the consequences of that state on different relevant aspects of a person's life, such as their physical function or social role, which are related to well-being or health-related quality of life [11].”

Although the article explores an interesting and important topic, the theoretical gaps and the theoretical basis for creating the scales are very limited. The introduction does not clearly indicate the gap and the need for the SPH scale. There is also a lack of systematisation of the clear differences and limitations of current approaches as well as current scales.

We greatly appreciate your comments. Therefore, we would like to inform you that we have completely restructured the introduction section in accordance with each of your observations, and we hope that we have complied with the corrections you have asked us to make.

The statistical analysis is adequate for the study. Too little detail is given on the choice of research samples and their justification, and it is not clear how many ranks the Likert scales had.

Dear reviewer, following your comment, which is similar to that of reviewer 2, we have detailed the reasons why we have chosen to use two different samples for the analyses. This new information can be found on lines 525-533.

“For this second sample, another group of participants who had not been involved in the first sample was selected. The reason for this was that, rather than seeking equivalence in the sample characteristics -such as age or sex-, what was intended was to confirm that the EFA results of the first sample would be stable in the application of a second sample totally independent of the first sample and achieve adequate results in the CFA [35]. It is also important to note that the reason the first group was not given the additional measurement instruments was because they were of no practical use during the EFA stage. However, since it was necessary to establish construct validity during the CFA stage, they were applied to the second sample of participants.”

Likewise, we would like to note that the required information regarding the response options is reported in lines 292-311 of the manuscript where the number of response options is described, and the reasons why an asymmetric likert scale without intermediate response was used. We are glad to reproduce this subsection:

“Six answer options on an asymmetric Likert-type scale were used for respondents to indicate their level of agreement with each item, where 1 was “Disagree”, 2 “Slightly agree”, 3 “Agree”, 4 “Mostly agree”, 5 “Strongly agree”, and 6 “Absolutely agree”. This format was used to increase the variability in the distribution of the data and prevent a ceiling effect, which is essential for psychometric procedures [30]. A six-point Likert scale, without a neutral point, was chosen to prevent the intermediate response bias [31] where people tend to use the intermediate option to avoid providing a compromised response to the items, which would affect the consistency of the participants' responses and consequently affect the performance of the exploratory and confirmatory factor analysis models. Additionally, we used asymmetric Likert-type response options because they are recommended when the intention is to measure variables where people tend to overestimate the perceptions that they have of themselves, causing a ceiling or floor effect [30]. Those authors have reported that using these types of response options allows researchers to obtain a wider variability in the participants' data, thus minimizing the skew and kurtosis statistics and allowing researchers to obtain a univariate and multivariate normal distribution, which are important elements for the proper analysis of the data.

For the case of self-perceived health, we consider it appropriate to use this response format since, in the context of the Mexican population, it has been reported that this population tends to make an erroneous self-perception of their own health due to the optimistic overestimation of health status [32].”

References are adequate for the study, but are very poor. The literature cited is not recent. There are no direct references to previous research on the topic and no measures or scales constructed to date.

As far as possible, we have tried to account for their accurate observation. However, due to the conceptual and operational confusion about self-perceived health, where it is often confused with quality of life, it was an arduous task to find recent literature that adequately uses this concept for quality studies. Therefore, the current references added were the following:

  • McEwen, B.S. Neurobiological and systemic effects of chronic stress. Chronic Stress (Thousand Oaks, Calif.) 2017, 1, 2470547017692328; DOI:10.1177/2470547017692328.
  • Heiestad, H.; Gjestvang, C.; Haakstad, L.A.H. Investigating self-perceived health and quality of life: A longitudinal prospective study among beginner recreational exercisers in a fitness club setting. BMJ Open 2020, 10, e036250; DOI:10.1136/bmjopen-2019-036250.
  • El Ansari, W.; Suominen, S.; Berg-Beckhoff, G. Is healthier nutrition behaviour associated with better self-reported health and less health complaints? Evidence from Turku, Finland. Nutrients 2015, 7, 8478–8490; DOI:10.3390/nu7105409.
  • Organization for Economic Cooperation and Development. Health at a Glance 2021: OECD Indicators. OECD Publishing Paris, 2021.
  • Kleinheksel, A.J.; Rockich-Winston, N.; Tawfik, H.; Wyatt, T.R. Demystifying Content Analysis. Am J Pharm Educ. 2020;84(1):7113. doi:10.5688/ajpe7113.
  • Ventura-León, J. De regreso a la validez basada en el contenido. Adicciones. 2019;34(4):1-4. Doi:20882/adicciones.1213.
  • Chyung, S.Y.Y.; Roberts, K.; Swanson, I.; Hankinson, A. Evidence-based survey design: The use of a midpoint on the likert scale. Perform. Improv. 2017, 56, 15–23; DOI:10.1002/pfi.21727.
  • Lloret-Segura, S.; Ferreres-Traver, A.; Hernández-Baeza, A.; Tomás-Marco, I. El análisis factorial exploratorio de los ítems: Una guía práctica, revisada y actualizada. An. Psicol. 2014, 30, 1151–1169; DOI:10.6018/analesps.30.3.199361.
  • Putnick, D.L.; Bornstein, M.H. Measurement invariance conventions and reporting: The state of the art and future directions for psychological research. Dev. Rev. DR 2016, 41, 71–90; DOI:10.1016/j.dr.2016.06.004.
  • Jarczok, M.N.; Kleber, M.E.; Koenig, J.; Loerbroks, A.; Herr, R.M.; Hoffmann, K.; Fischer, J.E.; Benyamini, Y.; Thayer, J.F. Investigating the associations of self-rated health: heart rate variability is more strongly associated than inflammatory and other frequently used biomarkers in a cross sectional occupational sample. PLoS One. 2015;10(2):1-19. doi: 10.1371/journal.pone.0117196.
  • Eddy, P.; Wertheim, E.H.; Hale, M.W.; Wright, B.J. Trait Mindfulness Helps Explain the Relationships Between Job Stress, Physiological Reactivity, and Self-Perceived Health. J Occup Environ Med. 2019;61(1):e12-e18. doi: 10.1097/JOM.0000000000001493.
  • Rodríguez, D. A.; Martínez-Cuervo, N.; Vázquez-Ortega, J. J.; Manjarrez-Ibarra, J.O; & Ríos-Quintero, Y. La temperatura nasal: marcador autonómico de relajación y su relación con el apoyo social en adultos mayores. Revista De PSICOLOGÍA DE LA SALUD. 2023;11(1):193-208. doi:10.21134/pssa.v11i1.319.

The Discussion section is quite limited. It needs to be significantly improved to highlight the results and their significance by explaining more clearly new or valuable points and results compared to previous studies. This section needs to include and discuss the results of the study with comments and support in agreement or disagreement with previous studies.

Estimated reviewer, we greatly appreciate your feedback about emphasizing the contributions of our results in the Discussion section. Therefore, we have described some points that we hope can meet your expectations regarding the improvement of our manuscript. This new information has been reported on lines 605-624. The text is reproduced below:

“The results obtained from the development and validation of the SPHS-12 can be considered of great value since, to date, the authors of this study had not been able to find another psychometric scale focused exclusively on the assessment of Self-perceived Health. As mentioned at the beginning of this paper, when assessing this construct, a single general question is usually used (“How good do you consider your health to be?” [18]), or questionnaires whose objectives are to assess other variables such as quality of life [15], which has led some authors to consider this variable as an imprecise measurement, despite its importance in the estimation of population mortality [51].

Although the Current Perceived Health-42 Questionnaire [52] was previously published, it cannot be considered as an SPH measurement instrument either, since the objective of this instrument was to assess the current state of health of the participants, which was considered by the authors as a different and independent variable from SPH. Thus, the SPHS-12 is probably the first scale specifically designed and validated to assess Self-Perceived Health in the general population.

Another advantage of this scale is its trifactorial configuration (Physical Health, Psychological Health, and Lifestyle), which allows for more specific assessments of what domains participants consider "Healthy", as opposed to other attempts to assess this perception through general questions that did not allow discerning whether what participants considered "healthy" was their cognitive capacity, their physical mobility, or their diet, to mention some aspects that are addressed in the items of the scale presented.”

Very laconically described Suggestions for Future Research section. Research limitations, theoretical and practical implications are missing.

Dear reviewer, we have responded to your suggestion to enrich the "Suggestions for future research" section. Here, we reproduce the added information from lines 684-695:

“Similarly, future works could opt to try to assess the criterion validity of SPHS-12 using biomarkers that reflect autonomic nervous system activity, such as peripheral temperature, Heart Rate Variability (HRV), respiratory rate, or inflammatory response using variables such as Interleukin-6 (IL-6), C-reactive protein, or Tumor Necrosis Factor alpha (TNF-α). The reasons for proposing these biomarkers are that recent studies have reported interesting associations where irregular values of these biomarkers (e.g., low peripheral temperature, high levels of IL-6, or sympathetic-dominant HRV) produce a low assessment of Self-perceived Health [54].

In this sense, also future research could evaluate the feasibility of using the SPHS-12 as an assessment tool in future psychological intervention protocols focused on fostering appropriate autonomic regulation through strategies such as mindfulness, diaphragmatic breathing, guided imagery, and progressive muscle relaxation [55,56].”

Reviewer 2 Report

I have reviewed the manuscript entitled "Design, Development, and Validation of the Self-Perceived 2 Health Scale (SPHS): Measurement of Patient-Reported Outcomes (PRO)" and would like to provide my feedback. Overall, I find the manuscript to be of high quality, with several strengths such as identifying a gap in the measurement of subjective health perception and proposing a new scale. The multi-stage development of the scale, including qualitative, inter-rater agreement, and factor analyses, is commendable. However, I believe there are areas where the authors can improve. I provide the following suggestions for further work:

  1. There appears to be overlap between the proposed measure and the construct commonly referred to as "health-related quality of life" (HRQoL). It would be beneficial to acknowledge and justify the need for the new scale in light of this alternative construct in the introduction.
  2. The authors should make a greater effort to ensure transparency in all phases of the research process. Specifically:

a. In phase 2.1.1, the authors should specify the literature consulted, the selection process, and the rationale for determining the three main dimensions (physical health, psychological health, and healthy lifestyle) based on the literature review. Additionally, they should include details about the material analyzed and the steps taken in the focus group analysis, providing a clear visualization of the analysis process and justifying the decisions made.

b. In phase 2.1.2, the authors should clarify the selection process for the experts involved and provide access to the database showing their answers. The purpose of the consultation with experts should be clearly stated, along with the methodology employed and the rationale behind the decisions made. It is also important to specify the formula used to calculate the Aiken's V index.

c. In phase 2.1.4, clarification is needed regarding the collection and analysis of concrete data from cognitive interviews. Additionally, the results obtained from the distribution of answers and the clarity of the items in the pilot test with 50 participants should be clearly stated.

d. In 3.2, the authors should provide complete outputs of the exploratory factor analysis, including the total loadings and their distribution in each factor for all successive versions obtained.

e. The entire scale as it was applied should be made accessible, enabling identification of the items corresponding to the different versions.

To accommodate space limitations, the authors may consider uploading this information in an appropriate open-access repository, with clear instructions provided in the text for accessing the supplementary information (see recommendations at: https://www.cambridge.org/core/journals/spanish-journal-of-psychology/article/how-do-you-behave-as-a-psychometrician-research-conduct-in-the-context-of-psychometric-research/3E0D34E46105C1E319E61E166F354579/share/4bd2477ae19882a37cb5de54070bc46f90552f75).

  1. In 2.2.1, it is unclear how the sample was subdivided and how the equivalence of the samples on relevant variables such as sex and age was ensured. The text suggests the presence of two independent samples, but this should be clarified. Additionally, the rationale behind providing the full version to one sub-sample and the reduced version to the other, as well as the inclusion of other measures in one sub-sample but not the other, needs clarification.
  2. The instruments section should include information on the fit of the measurement models assumed for each incorporated measurement, ensuring consistency with subsequent analyses.
  3. It is not appropriate to compare the fit of a model in one sample with other models estimated in a different sample. To make a valid comparison, the analysis should be conducted within a single sample.
  4. In section 3.5, the statement regarding the construct validity of the SPHS-12 scale should be revised. The results indicate a correlation between SPHS-12 and coping strategies evaluated using the COPE-28 scale, which is similar in magnitude to the BSI-18 (Physical Health) and even higher for the Lifestyle factor. This should be addressed in the manuscript.

I hope these suggestions prove helpful for the authors in revising their manuscript. Overall, the study has great potential, and addressing these points will further enhance the quality and clarity of the research.

Best regards,

Author Response

Comments and Suggestions for Authors:

I have reviewed the manuscript entitled "Design, Development, and Validation of the Self-Perceived 2 Health Scale (SPHS): Measurement of Patient-Reported Outcomes (PRO)" and would like to provide my feedback. Overall, I find the manuscript to be of high quality, with several strengths such as identifying a gap in the measurement of subjective health perception and proposing a new scale. The multi-stage development of the scale, including qualitative, inter-rater agreement, and factor analyses, is commendable. However, I believe there are areas where the authors can improve. I provide the following suggestions for further work:

There appears to be overlap between the proposed measure and the construct commonly referred to as "health-related quality of life" (HRQoL). It would be beneficial to acknowledge and justify the need for the new scale in light of this alternative construct in the introduction.

We appreciate the observation about the overlap between the constructs. Therefore, the need for a valid and reliable measurement scale that is both conceptually and operationally different from those used to assess HRQoL has been emphasized. This information is in the lines 157-166, and appears such as:

"In view of this conceptual and operational confusion, there is a need to be able to differentiate the construct of SPH from Health-Related Quality of Life, as the latter has been developed to assess those aspects of the individual's subjective experience focused primarily on how illness, disability, and treatment itself impacts on people's quality of life [20]. In contrast, the study of SPH has focused on identifying the functional physical status of individuals, regardless of whether any illness, disability, or ongoing treatment is present [21]. This becomes relevant since, as reported by Moon [20] in the absence of measurement instruments really focused on the assessment of SPH, the use of instruments designed to assess quality of life is considered appropriate, despite the differences that have been pointed out regarding the objectives of each instrument."

The authors should make a greater effort to ensure transparency in all phases of the research process. Specifically:

In phase 2.1.1, the authors should specify the literature consulted, the selection process, and the rationale for determining the three main dimensions (physical health, psychological health, and healthy lifestyle) based on the literature review.

We appreciate your pointing out the need to go deeper into the reasons behind the selection of these factors. We have therefore added the following lines in the manuscript (Lines 228-235):

“The reasoning behind determining Physical Health, Psychological Health and Healthy Lifestyle as the main dimensions, and therefore conducting the focus groups based on these factors, was to provide conceptual coherence to the construct of this study since, as Moon [20] reports, frequently, although the aim of the study is to assess Self-perceived Health, measurement instruments and definitions derived directly from the Quality of Life construct are often used. Therefore, this decision was made based on the contributions of different authors who have worked on SPH [1,2,3,6,11].”

Additionally, they should include details about the material analyzed and the steps taken in the focus group analysis, providing a clear visualization of the analysis process and justifying the decisions made.

Following the previous point, we have included details on the procedures carried out to synthesize the information obtained in the focus groups, as detailed in the lines 240 – 254 of the manuscript:

" To make sense of the transcribed text, a content analysis was performed where, among everything reported by the participants, meaning units that were related to the three central factors were first identified and labeled according to the factor to which they belonged in order to give greater structure to the information collected [27]. For example, some details of the material analyzed were phrases about the activities carried out by the participants that reflected adequate physical health such as "Frequency of mobility, not being static constantly. Being under constant movement", "Balance with the way you self-regulate, cope with emotions, a loss, or some discontent around the day" regarding psychological health, and "having adequate habits at work and rest; considering that being every day working is not totally good, since relaxation and rest are part of health" referring to Healthy Lifestyle. This allowed us to identify the themes that described behaviors, experiences or emotions experienced by the participants with respect to the questions asked, offering a more detailed description that allowed us to made 52 items and expand the number of factors to consider those that Shields & Shooshtari [13], and Jylhä [2] recommended:...”

In phase 2.1.2, the authors should clarify the selection process for the experts involved and provide access to the database showing their answers. The purpose of the consultation with experts should be clearly stated, along with the methodology employed and the rationale behind the decisions made. It is also important to specify the formula used to calculate the Aiken's V index.

The selection process of the experts involved has been detailed, as well as the objective of their evaluation. The formula for calculating the Aiken's V index has also been added.

Also, as you suggested, we have made public the expert evaluation database through the Mendeley repository, which can be accessed through the link https://doi.org/10.17632/8wrysjbsny.1.

The changes made are on the following lines 273-286:

“The objective of conducting this expert evaluation was to assess the degree of relevance among the items to support the proposed conceptual definition, and the degree to which the number of items was sufficient to adequately represent their respective factor. To this end, the experts who agreed to participate were given an e-mail form asking them to evaluate on a scale of 1 (Not at all relevant) to 5 (Fully relevant) each of the items; and on a scale of 1 (Not at all represented) to 5 (Fully represented) the degree to which the items were sufficient to support their respective factors. This was done as we sought to ensure that the final items were sufficiently consistent with the proposed conceptual definition, while achieving adequate conceptual coverage of their factors [29]. The formula used to calculate the Aiken's V index was  where XÌ… is the mean of the judges' evaluation scores, l is the lowest score that is possible to obtain, and k is the difference between the highest and lowest score on the rating scale [28, 29]. The database with the experts' evaluations can be accessed at the following link https://doi.org/10.17632/8wrysjbsny.1.”

In phase 2.1.4, clarification is needed regarding the collection and analysis of concrete data from cognitive interviews (Lines 223 – 238). Additionally, the results obtained from the distribution of answers and the clarity of the items in the pilot test with 50 participants should be clearly stated.

It is hoped that the required information about the collection and analysis of both the cognitive interviews has been clarified in detail (lines 315 – 330):

“The cognitive interviews were conducted face-to-face following the recommendations of Willis [34], where one interview was conducted using the "Thinking aloud" technique, asking the participant to verbalize everything that came to mind while answering the scale. The second interview was conducted using a "Concurrent Paraphrasing" technique where the participant was asked to say the items in his or her own words as he or she answered the scale. The third interview was conducted using a "Retrospective paraphrasing" technique, where the participant, once he/she had completed the scale, was asked to state as many of the items as he/she could remember. The fourth interview was conducted by means of "exhaustive paraphrasing", where the participant was asked to restate in his/her own words all the items of the scale (items, response options and instructions). The fifth interview was conducted using the "Selective paraphrasing" technique, where the participant rephrased in his/her own words those items that, while answering the scale, he/she considered the most confusing. The objective of these interviews was to identify if there were items, answer options or instructions that were unfamiliar to the participants. Since the interviewees correctly paraphrased all the items on the scale, there was no need to modify them.”

As well as the distribution of answers and the clarity of the items in the pilot test with 50 participants (lines 334-338):

“the distribution obtained from the application of the 50 participants since their skewness and kurtosis were < |2|, the means obtained (lower XÌ… = 3.08, upper XÌ… = 5.06) were close to the theoretical mean (XÌ… = 3.5, for six response options), and at the time of concluding the application, no participant reported having incomprehensible items when asked about the clarity of the scale.”

  1. In 3.2, the authors should provide complete outputs of the exploratory factor analysis, including the total loadings and their distribution in each factor for all successive versions obtained.

  1. The entire scale as it was applied should be made accessible, enabling identification of the items corresponding to the different versions.

To accommodate space limitations, the authors may consider uploading this information in an appropriate open-access repository, with clear instructions provided in the text for accessing the supplementary information (see recommendations at: https://www.cambridge.org/core/journals/spanish-journal-of-psychology/article/how-do-you-behave-as-a-psychometrician-research-conduct-in-the-context-of-psychometric-research/3E0D34E46105C1E319E61E166F354579/share/4bd2477ae19882a37cb5de54070bc46f90552f75).

Taking into consideration both point d and e, the requested information (factorial loading, items distributions and communalities) of EFA of all the successive versions obtained, i.e., the 50-item, 38-item, 30-item, 24-item & 18-item scales have been incorporated into the manuscript in table 1. The 12-item scale is reported in Table 2, as it was previously presented.

In 2.2.1, it is unclear how the sample was subdivided and how the equivalence of the samples on relevant variables such as sex and age was ensured. The text suggests the presence of two independent samples, but this should be clarified. Additionally, the rationale behind providing the full version to one sub-sample and the reduced version to the other, as well as the inclusion of other measures in one sub-sample but not the other, needs clarification.

Thanks to your comments, we have discussed in more detail the reasons for using a second sample of participants on the lines 525-533:

“For this second sample, another group of participants who had not been involved in the first sample was selected. The reason for this was that, rather than seeking equivalence in the sample characteristics -such as age or sex-, what was intended was to confirm that the EFA results of the first sample would be stable in the application of a second sample totally independent of the first sample and achieve adequate results in the CFA [35]. It is also important to note that the reason the first group was not given the additional measurement instruments was because they were of no practical use during the EFA stage. However, since it was necessary to establish construct validity during the CFA stage, they were applied to the second sample of participants.”

The instruments section should include information on the fit of the measurement models assumed for each incorporated measurement, ensuring consistency with subsequent analyses.

Fit indices were added for the measurement scales used.

The Subjective Well-being scale (lines 303-304) presented a CMIN/DF = 2.45, CFI = .996, GFI = .993 & RMSEA = .046.

The COPE-28 (lines 313-314) presented a CMIN/DF = 1.699, CFI = .957, GFI = .906 & RMSEA = .048, according to the model originally proposed by the authors.

The brief symptom inventory (BSI-18) a CMIN/DF = 1.13, CFI = .996, GFI = .978 & RMSEA = .026 were reported on lines 325-326.

It is not appropriate to compare the fit of a model in one sample with other models estimated in a different sample. To make a valid comparison, the analysis should be conducted within a single sample.

We considered your valuable observation, so in Table 4, we have added the model fits of the 12-item version of the scale applied to the first sample (n = 303), noting that, even if they were samples of independent participants, their fit indices do not differ substantially.

In section 3.5, the statement regarding the construct validity of the SPHS-12 scale should be revised. The results indicate a correlation between SPHS-12 and coping strategies evaluated using the COPE-28 scale, which is similar in magnitude to the BSI-18 (Physical Health) and even higher for the Lifestyle factor. This should be addressed in the manuscript.

Thank you very much for making this observation. In the manuscript we have stated the following in lines 590-595:

"Based on these data, it is suggested that the construct validity of the SPHS-12 was adequate for convergent (SWBS-8) and divergent (BSI-18) validity. However, the discriminant validity of the SPHS-12 was barely sufficient given that, although the correlations between the COPE-28 and SPHS-12 were the lowest when compared to the BSI-18 & SWBS-8, the correlation between COPE-28 and lifestyle was higher than between BSI-18 and lifestyle.”

I hope these suggestions prove helpful for the authors in revising their manuscript. Overall, the study has great potential, and addressing these points will further enhance the quality and clarity of the research.

Round 2

Reviewer 2 Report

Thank you very much for listening to my suggestions. I consider that the following points should be revised again:

- "In 3.2, the authors should provide the full results of the exploratory factor analysis, including the total loadings and their distribution on each factor for all successive versions obtained." Here I meant to put the loadings that were obtained for all factors on the given items. The tables only show the loadings for the factor to which the item theoretically belongs but omit the remaining loadings.

- "The entire scale as it was applied should be accessible, allowing the identification of the items corresponding to the different versions." Here I meant providing access to the instrument as it was applied to the participants.

- Finally, I suggest providing access to other important research materials, particularly the syntaxes used and steps followed in each software used, as well as the database on which the factor analysis was based.

Author Response

Comments and Suggestions for Authors

Thank you very much for listening to my suggestions. I consider that the following points should be revised again:

- "In 3.2, the authors should provide the full results of the exploratory factor analysis, including the total loadings and their distribution on each factor for all successive versions obtained." Here I meant to put the loadings that were obtained for all factors on the given items. The tables only show the loadings for the factor to which the item theoretically belongs but omit the remaining loadings.

Esteemed reviewer:

We thank you very much for your effort in clarifying us your past observation. It seems that we have been able to add the requested information in Table 1 and Table 2 where the factor loadings of the items obtained are shown, regardless of whether they achieved the cut-off points established in the analyses.

- "The entire scale as it was applied should be accessible, allowing the identification of the items corresponding to the different versions." Here I meant providing access to the instrument as it was applied to the participants.

Thank you very much for your clarification, esteemed reviewer.

In response to your comment, it has been decided to upload all the instruments applied as supplementary material to the article so that they can be freely available to anyone interested.

- Finally, I suggest providing access to other important research materials, particularly the syntaxes used and steps followed in each software used, as well as the database on which the factor analysis was based.

Likewise, dear reviewer, it has been decided to upload all the databases used, as well as the R syntax, as supplementary material to the article. In addition, a specific note has been prepared explaining how to proceed to perform the analyses reported in the study, and a contact email to answer any questions that may arise regarding the procedure performed.